# Polyploidy, regular patterning of genome copies, and unusual control of DNA partitioning in the Lyme disease spirochete

Constantin N. Takacs [1,2,3], Jenny Wachter [4,9,10], Yingjie Xiang[5,6,10], Zhongqing Ren[7,10], Xheni Karaboja[7], Molly Scott[6,8], Matthew R. Stoner [3,6,8], Irnov Irnov[1,2,3], Nicholas Jannetty[3,6,8], Patricia A. Rosa[4], Xindan Wang [7] ✉ & Christine Jacobs-Wagner [1,2,3] ✉

*Borrelia burgdorferi*, the tick-transmitted spirochete agent of Lyme disease, has a highly segmented genome with a linear chromosome and various linear or circular plasmids. Here, by imaging several chromosomal loci and 16 distinct plasmids, we show that *B. burgdorferi* is polyploid during growth in culture and that the number of genome copies decreases during stationary phase. *B. burgdorferi* is also polyploid inside fed ticks and chromosome copies are regularly spaced along the spirochete's length in both growing cultures and ticks. This patterning involves the conserved DNA partitioning protein ParA whose localization is controlled by a potentially phage-derived protein, ParZ, instead of its usual partner ParB. ParZ binds its own coding region and acts as a centromere-binding protein. While ParA works with ParZ, ParB controls the localization of the condensin, SMC. Together, the ParA/ParZ and ParB/SMC pairs ensure faithful chromosome inheritance. Our findings underscore the plasticity of cellular functions, even those as fundamental as chromosome segregation.

Lyme disease is the most prevalent vector-borne infectious disease in North America and Europe[1]. Its geographic range has steadily spread over the years, with caseloads recently estimated to be near 500,000 per year in the United States[1,2]. Lyme disease is caused by *Borrelia burgdorferi* and related spirochete bacteria[3]. In nature, Lyme disease spirochetes undergo a transmission cycle between *Ixodes* hard tick vectors and warm-blooded vertebrate hosts[4]. Infection in humans via a tick bite can result in a wide variety of symptoms when left untreated. Disease manifestations range from skin rashes, fever, and malaise during early stages of the disease to arthritis, carditis, and neurological

symptoms during later stages[3]. Given *B. burgdorferi*'s medical relevance, we set out to study basic biological processes necessary for cell proliferation. This topic is of considerable interest because bacterial multiplication is a prerequisite for successful transmission, host infection, and disease causation.

*B. burgdorferi* was identified in 1982[5]. Despite four decades of research, many of the fundamental cellular processes underlying the ability of this bacterium to self-replicate remain understudied. This is in part because *B. burgdorferi* has a long doubling time (5 to 18 h) in culture[6–11]. Additionally, genetic manipulation of this organism, while

[1]Department of Biology, Stanford University, Palo Alto, CA, USA. [2]Sarafan ChEM-H Institute, Stanford University, Palo Alto, CA, USA. [3]The Howard Hughes Medical Institute, Palo Alto, CA, USA. [4]Laboratory of Bacteriology, Rocky Mountain Laboratories, Division of Intramural Research, National Institute of Allergy and Infectious Diseases, National Institutes of Health, Hamilton, MT, USA. [5]Department of Mechanical Engineering, Yale University, New Haven, CT, USA. [6]Microbial Sciences Institute, Yale West Campus, West Haven, CT, USA. [7]Department of Biology, Indiana University, Bloomington, IN, USA. [8]Department of Molecular, Cellular, and Developmental Biology, Yale University, New Haven, CT, USA. [9]Present address: Bacterial Vaccine Development Group, Vaccine and Infectious Disease Organization, University of Saskatchewan, Saskatoon, SK, Canada. [10]These authors contributed equally: Jenny Wachter, Yingjie Xiang, Zhongqing Ren. ✉e-mail: xindan@indiana.edu; jacobs-wagner@stanford.edu

possible[12–14], remains challenging compared to that of *Escherichia coli* and other common model bacteria. However, knowledge obtained from the study of model bacteria does not always translate to unrelated species, including spirochetes. These bacteria form a phylum of particular interest because, in addition to *B. burgdorferi*, it includes important human pathogens such as the agents of relapsing fever, syphilis, and leptospirosis.

In this study, we focused on one of the most enigmatic and important aspects of *B. burgdorferi's* biology: genome inheritance. Faithful genome inheritance during cellular replication is essential for the propagation of all life forms. Despite its small size of ~1.5 megabases[15], the *B. burgdorferi* genome is the most segmented bacterial genome known to date[16]. It is composed of one linear chromosome and over 20 linear or circular plasmids, several of which have essential roles during the spirochete's natural tick-vertebrate transmission cycle[4,15,17,18]. Quantitative polymerase chain reaction (qPCR)-based measurements[19] generated the common (though not universal[20]) view that *B. burgdorferi* has about one chromosome copy per cell. Endogenous plasmids have between one and three times the copy number of the chromosome[21–25]. Given the large size of *B. burgdorferi* cells (10 to 25 μm or longer depending on the strain)[26–28], monoploidy means that the replicated chromosome and plasmids would have to segregate over long distances to ensure their faithful inheritance during division. This, however, has not been examined experimentally.

Experiments in a heterologous *E. coli* system, together with a transposon screen in *B. burgdorferi*, have suggested that the replication and partitioning of *B. burgdorferi* plasmids are mediated by specific plasmid-encoded proteins via unknown mechanisms[18,29–31]. On the other hand, chromosome segregation is predicted to involve a ParA/ParB system[15,32]. ParA and ParB proteins are well known to work together to mediate the segregation of duplicated chromosomal origins of replication (*oriC*) in broadly diverse bacteria[32,33]. ParB is often referred to as a "centromere-binding" protein because it specifically binds to centromere-like sequences (*parS*) usually located near *oriC*[32]. After loading on the DNA at the *parS* sites, ParB spreads onto adjacent sequences[34–36] to form a partition complex. ParA is an ATPase that dimerizes upon ATP binding, which in turn promotes the nonspecific binding of the ParA dimer to the DNA (the nucleoid)[37,38]. Upon interaction with ParA, the ParB-rich partition complex stimulates the ATPase activity of ParA, causing dimer dissociation and release of ParA from the DNA[38,39]. Repetition of this biochemical cycle, combined with a translocation force[40] derived from the elastic properties of the chromosome[41–44] and/or a diffusion-based mechanism[45–48], drives the translocation and therefore the segregation of replicated partition complexes[33,49]. The *B. burgdorferi* chromosome contains a *parS* site[32] near *oriC* and encodes both ParA and ParB homologs[15], predictive of a conserved ParA/ParB function in chromosome segregation.

In this study, we genetically labeled and imaged various chromosomal loci and plasmids in live *B. burgdorferi* cells. Fluorescence microscopy analysis, combined with genetic deletions and ChIP-seq experiments, revealed that *B. burgdorferi* is polyploid and uses a novel centromere-binding protein, rather than ParB, to carry out a key DNA partitioning activity.

## Results

### *B. burgdorferi* cells carry multiple chromosome copies during growth

To label *oriC* in live *B. burgdorferi* cells, we relied on the specific recognition of the *oriC*-proximal *parS* site by ParB (see Supplementary Notes and Supplementary Fig. 1a, b). We therefore substituted the *parB* gene with *mcherry-parB* at the endogenous locus. The resulting strain (CJW_Bb474), which also expressed free GFP for cytoplasm visualization, was stained with the Hoechst dye to reveal DNA localization. In this strain, we would expect one or two fluorescent mCherry-ParB

puncta within the nucleoid to reflect a monoploid state before and after initiation of chromosomal DNA replication, respectively. However, each cell had multiple mCherry-ParB foci that appeared regularly spaced along the nucleoid (Fig. 1a).

We confirmed this finding in 13 additional *B. burgdorferi* strains derived from different lineages of the strain B31, or from other *B. burgdorferi* isolates, including the well-studied N40 and 297 strains (Fig. 1b and Supplementary Fig. 1c; also see Supplementary Notes for strain construction and *oriC* labeling). The spacing between adjacent *oriC* copies was similar across all tested strains (Fig. 1b). As a complementary approach, we labeled the *oriC*-proximal *uvrC* locus of the *B. burgdorferi* B31 chromosome with the orthogonal msfGFP-ParB[P1]/ *parS*[P1] pair derived from the *E. coli* P1 plasmid and observed similar results (see Supplementary Notes and Supplementary Fig. 1a, b, d–f). To demonstrate that the detection of multiple *oriC* copies reflects the presence of multiple chromosomes per cell, we genetically labeled and visualized the left and right telomeres (*terC*) using the msfGFP-ParB[P1]/ *parS*[P1] system (Fig. 1c–e). We also confirmed the presence of multiple *terC* copies per cell by DNA fluorescence in situ hybridization (Supplementary Fig. 1g). Collectively, our data show that *B. burgdorferi* cells contain multiple complete chromosomes and thus are polyploid during exponential growth in culture.

We found that the discrepancy between our polyploidy results and previous qPCR measurements of ~1.3 chromosomes per cell[19] stemmed from a difference in the culture growth stage. While we imaged cultures in exponential growth (Fig. 1a–e, Supplementary Fig. 1b–g), the previous analysis was done using saturated cultures[19], which had likely reached stationary phase. Indeed, we found that the *oriC* copy number decreases in stationary phase cultures, ultimately reaching about one copy per cell (Fig. 1f, g). qPCR measurements of chromosomal copies in different growth stages confirmed that exponentially growing cells contain multiple chromosome copies and that their copy numbers decrease in the stationary phase (Supplementary Fig. 1h), in agreement with our imaging results (Fig. 1f, g). In fact, these findings are conceptually consistent with a previous study by Ornstein and Barbour[20].

### *B. burgdorferi* cells contain multiple copies of their endogenous plasmids

Next, we examined the localization and copy number of 16 distinct plasmids relative to the chromosome by generating different strains, each with a distinct endogenous plasmid labeled in addition to the *oriC* locus (Fig. 2a, Supplementary Data 1). For each plasmid, we detected multiple copies per cell (Fig. 2a, Supplementary Fig. 2a). The plasmid:*oriC* ratios varied between ~0.5 and ~1.8 (Fig. 2b, Supplementary Fig. 2a), which overlaps with previous findings that documented plasmid to chromosome copy number ratios between 1:1 and 3:1[21–23,25]. These results were further confirmed by marker frequency analysis using whole genome sequencing (see methods) (Fig. 2b). Moreover, the plasmids were regularly spaced within the cells (Fig. 2a). As with the chromosome, the copy number of cp26 (Supplementary Fig. 2b), and presumably that of the other plasmids, decreased as the axenic culture advanced into stationary phase, suggesting a potential coordination between independent segments of the genome.

### *B. burgdorferi* cells in fed nymphs are also polyploid

While the phenotypes of stationary phase cultures were interesting (and will be explored in a separate study), we chose to focus our attention on *B. burgdorferi* during growth because cell proliferation is required for disease to occur. Outside the laboratory, *B. burgdorferi* cannot grow as a free-living organism. Therefore, to test whether the polyploidy observed in growing cultures has physiological relevance in a natural context, we colonized ticks with strain CJW_Bb474 expressing mCherry-ParB and cytoplasmic GFP through feeding on infected mice. This strain displayed no apparent defect in mouse infectivity either by

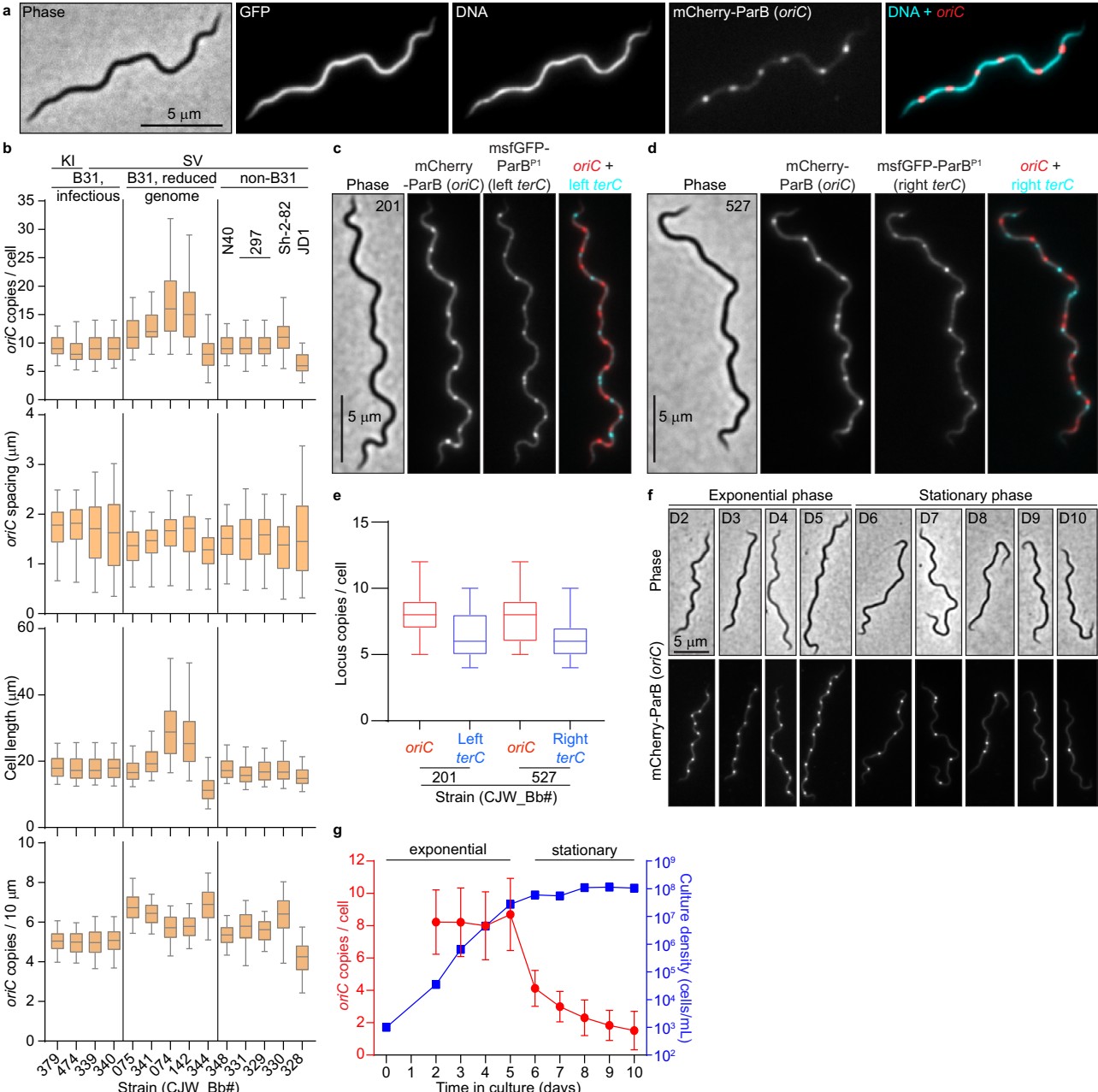

**Fig. 1 | *B. burgdorferi* cells contain multiple chromosome copies. a** Images of a cell of strain CJW_Bb474 expressing cytosolic free GFP and mCherry-ParB. Hoechst 33342 was used to stain the DNA. **b** Population quantifications of exponentially growing cultures of the strains shown in Supplementary Fig. 1c. Top to bottom: copy numbers per cell of the labeled *oriC* loci; *oriC* spacing (distances between adjacent *oriC* spots); cell lengths; and *oriC* densities (copies per 10 μm of cell length). Strain numbers are indicated at the bottom. Shown are the mean of the data (middle line), the 25 to 75 percentiles of the data (box), and the 2.5 to 97.5 percentiles of the data (whiskers). Strain backgrounds and *oriC* labeling method (*mcherry-parB* knock-in, KI, or tagged *parB* expressed from a shuttle vector, SV) are listed at the top. **c** Phase-contrast and fluorescence micrographs of a cell of strain CJW_Bb201, which expresses labels for *oriC* (mCherry-ParB) and the left *terC* locus (msfGFP-ParB^P1), respectively. **d** Phase-contrast and fluorescence micrographs of a cell of strain CJW_Bb527, which expresses labels for *oriC* (mCherry-ParB) and the

right *terC* locus (msfGFP-ParB^P1), respectively. d and e. Images were acquired while the cultures were growing exponentially. **e** Boxplots of *oriC* and *terC* copies per cell in exponentially growing cultures of strains CJW_Bb201 and CJW_Bb527. Shown are the mean of the data (middle line), the 25 to 75 percentiles of the data (box) and the 2.5 to 97.5 percentiles of the data (whiskers). **f** An exponentially growing culture of strain CJW_Bb339, in which *oriC* is labeled by expression of mCherry-ParB, was diluted to 10³ cells/mL then imaged daily from day 2 through day 10 of growth in culture. Shown are representative images of cells from the days indicated on the phase-contrast images. **g** Plot showing the *oriC* copy number per cell (red, mean ± standard deviation) and the culture density (blue, in cells/mL) over time for the population imaged in (**f**). Source data for panels (**b**, **e**, and **g**) are provided as a Source Data file. The numbers (*n*) of cells analyzed and the number of replicates are provided in Supplementary Data 2.

needle injection or tick bite, or in acquisition by ticks feeding on infected mice (Supplementary Fig. 1i–k and Supplementary Notes). Using three-dimensional deconvolution of image stacks, we were able to readily detect GFP-positive spirochetes in the midgut of fed, CJW_Bb474-colonized nymphs (Fig. 3). These spirochetes, which were

co-stained with the DNA dye Hoechst, contained regularly spaced mCherry-ParB foci (Fig. 3). The complex three-dimensional orientation of the thin spirochetes within the tick midgut[50] prevented us from accurately determining the number of *oriC* copies per cell or measuring distances between adjacent *oriC* copies. Nevertheless, based on

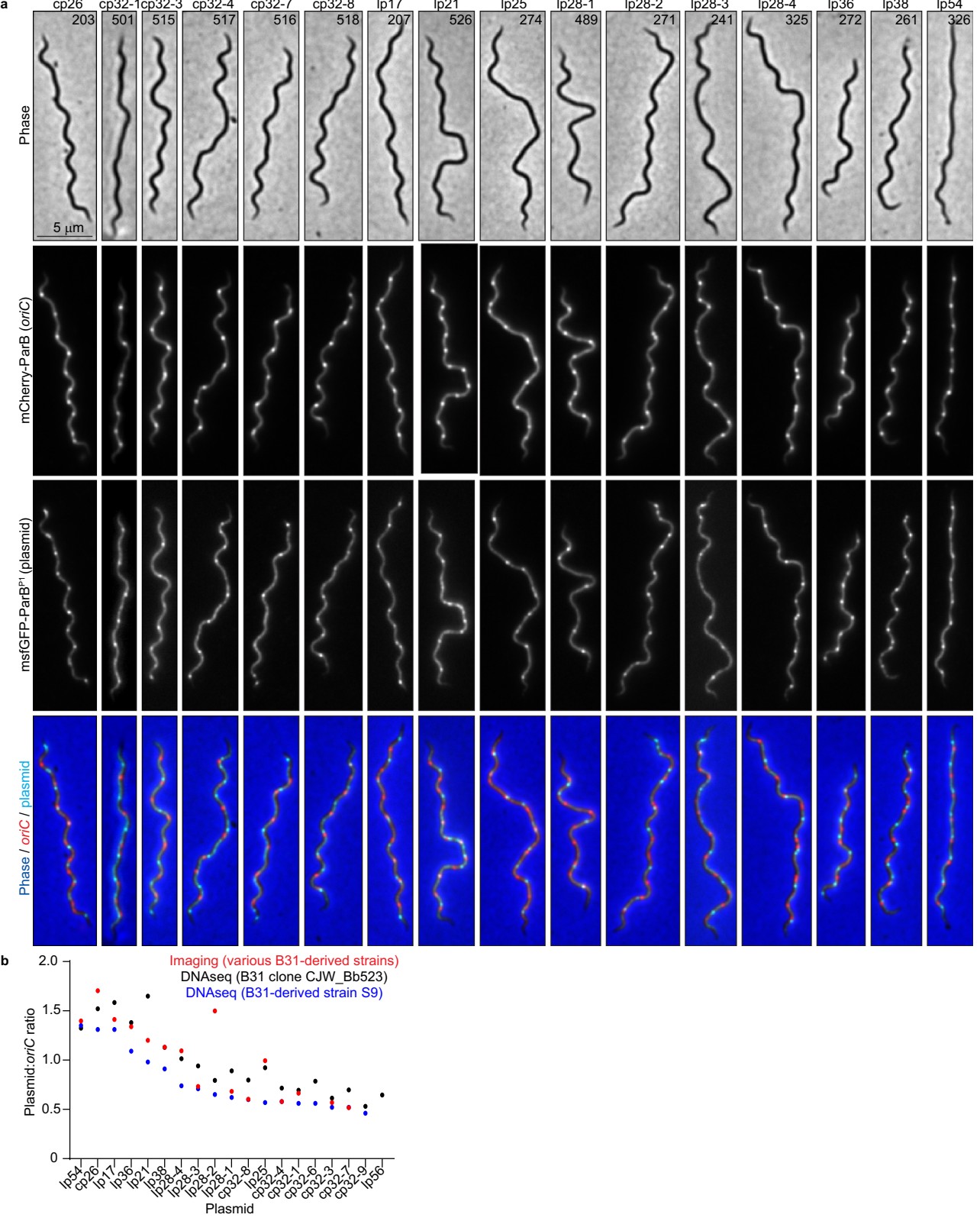

**Fig. 2 | Exponentially growing *B. burgdorferi* cultures contain multiple copies per cell of their endogenous plasmids. a** Images of representative cells of strains in which the *oriC* loci were labeled by expression of mCherry-ParB while the plasmid indicated at the top was labeled by insertion of a *parS*^P1 sequence and expression of msfGFP-ParB^P1. CJW_Bb strain numbers are marked on the phase-contrast images. **b** Plot showing the plasmid-to-*oriC* copy number ratios determined by imaging the

strains shown in (**a**) (red), or by marker frequency analysis of whole genome sequencing data (see methods) in strains S9 (blue) or strain CJW_Bb523, a clone of strain B31MI (black). Source data are provided as a Source Data file. The numbers (*n*) of cells analyzed and the number of replicates are provided in Supplementary Data 2.

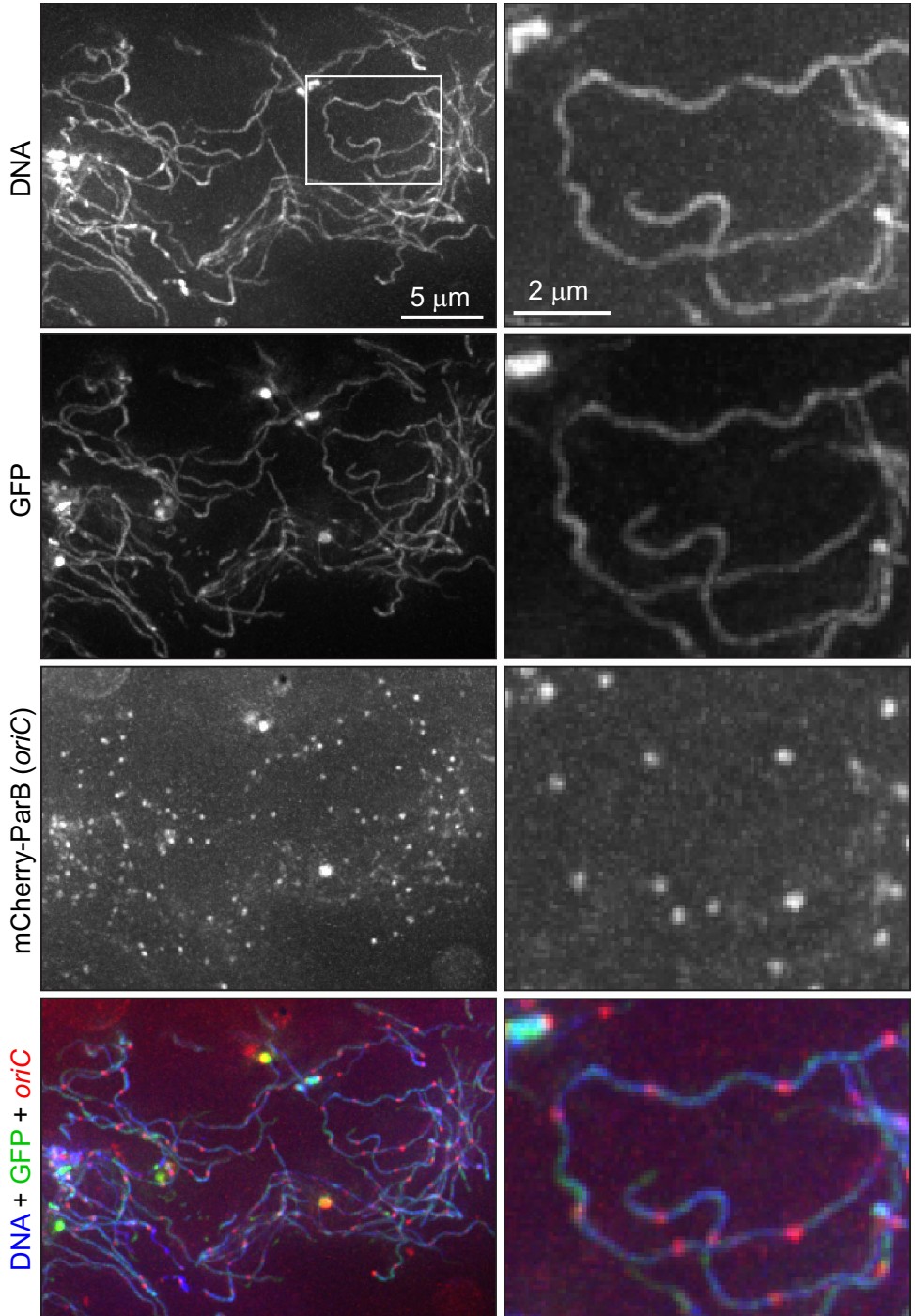

**Fig. 3 | Tick-borne *B. burgdorferi* cells contain multiple chromosome copies.** Images of cells of strain CJW_Bb474 in the midgut of a nymphal tick 10 days after feeding drop-off. DNA was stained with Hoechst 33342. The mCherry-ParB *oriC* signal was amplified by staining with RFP booster. Shown are max-Z projections of a deconvolved stack of images (left) and higher-magnification views of the region indicated by the white rectangle (right).

the inset in Fig. 3, we estimated the inter-origin distance to be ~2 μm, which is only slightly larger than the distances measured in growing cultures (Fig. 1b). Importantly, we establish that the polyploidy of *B. burgdorferi* cells is physiologically relevant.

**B. burgdorferi oriC copy numbers correlate with cell length**
In culture, strains with longer cells had higher *oriC* copy numbers per cell (Fig. 1b). Correlation between *oriC* copy number and cell length was also apparent at the single-cell level for each strain (Supplementary Fig. 3a, b), a scaling property that persisted after blocking cell

division using the FtsI inhibitor piperacillin[51] (Fig. 4a, b). The number of *terC* copies also correlated linearly with cell length (Fig. 4c, d), as did the number of plasmid copies (Supplementary Fig. 3c, d). We therefore approximated chromosome and plasmid densities by calculating the number of copies found in 10 μm of cell length (Fig. 1b and Supplementary Fig. 2a). Among the B31-derived strains, those with reduced genomes (i.e., having fewer endogenous plasmids) had higher *oriC* densities than those with more endogenous plasmids (Fig. 1b), suggesting that *B. burgdorferi* may initiate DNA replication in response to the cellular space available to be filled by the DNA.

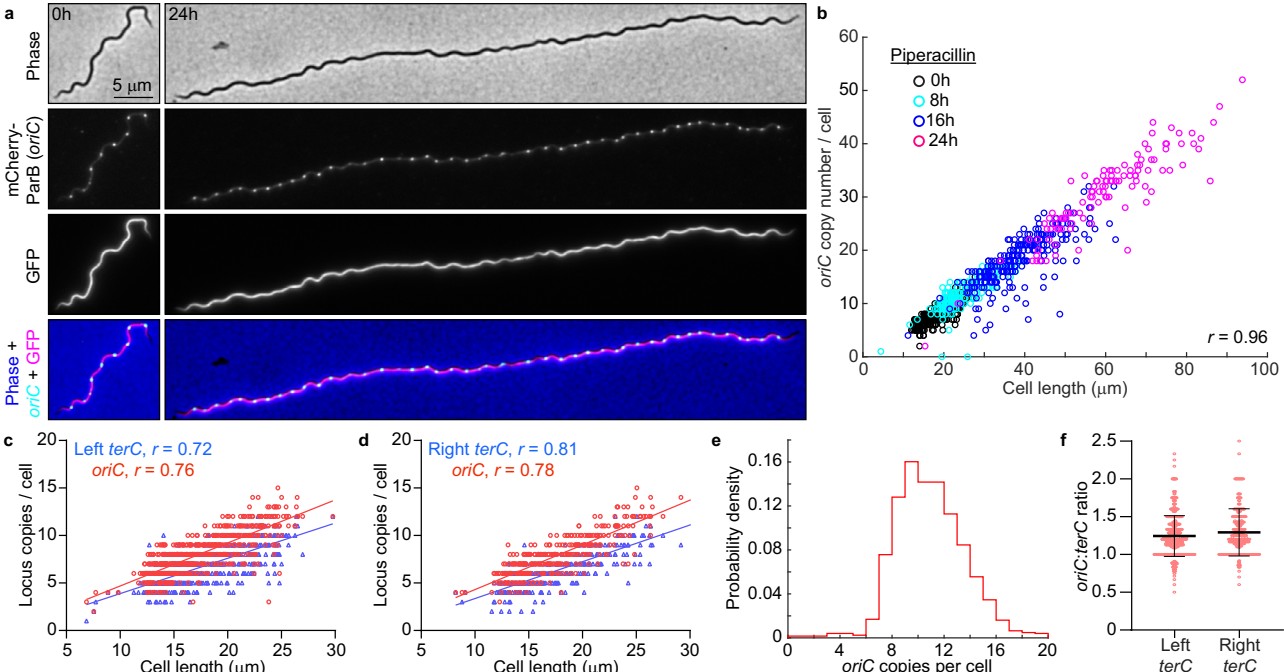

**Fig. 4 | *B. burgdorferi* chromosome copy numbers correlate with cell length.**
**a** Images of cells of strain CJW_Bb474 before (0 h) or after 24 h of exposure to piperacillin. **b** Correlation of *oriC* copy number per cell with cell length in strains CJW_Bb474 and CJW_Bb379 following the indicated durations of piperacillin treatment. *r*, Pearson's correlation coefficient. **c** Correlations of *oriC* and *terC* copy number per cell with cell length in exponentially growing cultures of strain CJW_Bb201. *r*, Pearson's correlation coefficient. **d** Correlations of *oriC* and *terC* copy number per cell with cell length in exponentially growing cultures of strain CJW_Bb527. *r*, Pearson's correlation coefficient. **e** Histogram of *oriC* copy numbers per cell in exponentially growing cultures of strain CJW_Bb379. **f** Plot showing the *oriC:terC* ratios calculated in single cells from exponentially growing cultures of strains CJW_Bb201 and CJW_Bb527, following fluorescent spot detection. Black lines depict means ± standard deviations. Source data for panels b-f are provided as a Source Data file. The numbers (*n*) of cells analyzed and the number of replicates are provided in Supplementary Data 2.

Several lines of evidence suggest that *B. burgdorferi* initiates DNA replication asynchronously. First, the *oriC* copy number in exponentially growing *B. burgdorferi* cultures had a unimodal distribution (Fig. 4e), as opposed to the expected bimodal distribution if all *oriC* copies would replicate at the same time. Second, at the population level, the *oriC:terC* ratio was ~1.2 on average, as measured both by imaging and by marker frequency analysis. Such an average *oriC:terC* ratio close to 1 indicates that few chromosomes are replicating at the same time, which we also confirmed at the single-cell level, as most individual cells had *oriC:terC* ratios smaller than 2 (Fig. 4f). Asynchronous chromosome replication and scaling between genome copy and cell length may be a common property of polyploid bacteria, as they are also observed in cyanobacteria[52,53].

## *B. burgdorferi* ParA is depleted at *oriC* loci

We next investigated the regular spacing of chromosome copies along the cell length, which ensures near-even distribution of the chromosome to daughter cells following division at midcell. Based on the known functions of ParA and ParB in other bacteria[33,49,54], we suspected that the *B. burgdorferi parA* and *parB* genes, which reside close to the predicted *parS* site (Fig. 5a), played an important role in the regular patterning of *oriC* copies in *B. burgdorferi*. As expected, we found that the formation of mCherry-ParB foci was dependent on the presence of *parS* (Supplementary Fig. 1b and Supplementary Notes). The genome-wide binding profile of mCherry-ParB, determined by ChIP-seq, revealed a broad enrichment peak that included the *parS* site (Fig. 6 and Supplementary Fig. 4a). Furthermore, chromosomal replacement of *parA* with *parA-msfgfp* revealed an uneven ParA-msfGFP signal distribution along the length of the cell (Fig. 7a). The regions of concentrated ParA-msfGFP signal alternated with regions of signal depletion that corresponded to the location of the mCherry-ParB foci in these cells, creating a banded localization pattern (Fig. 7a–c). While

ParA-msfGFP displayed modest accumulation between mCherry-ParB foci that were in close proximity, it accumulated prominently between mCherry-ParB foci that were farther apart (Fig. 7b, c). This localization pattern is expected for ParA/ParB DNA partitioning systems[42,43], as ParB is known to stimulate ParA depletion[55,56].

## ParZ, not ParB, controls ParA localization in *B. burgdorferi*

As ParB normally controls ParA localization in other bacterial ParA/ParB systems[57,58], we anticipated that deletion of *parB* and *parS* (*parBS*) would disrupt the ParA banded pattern and result in a more uniform distribution. Surprisingly, deleting *parBS* did not eliminate the banded ParA-msfGFP localization in *B. burgdorferi* (Fig. 7d). Seeking an explanation for this phenotype, we inspected the *B. burgdorferi par* system more closely. We noticed that *B. burgdorferi* ParB lacked an otherwise conserved N-terminal peptide (Fig. 5b, Supplementary Fig. 5a), which is required for stimulation of ParA ATPase activity in other ParA/ParB systems[38,39]. The absence of this peptide from *B. burgdorferi* ParB explains its inability to control ParA localization. The ParB proteins of staphylococci and streptococci also lacked this N-terminal peptide (Fig. 5b, Supplementary Fig. 5a). However, these bacteria also lack a ParA homolog[32,59,60] and thus do not need the ParA control function of ParB. The rest of the *B. burgdorferi* ParB sequence is similar to that of other chromosomal ParB proteins (Supplementary Fig. 5a). *B. burgdorferi* ParA also has a typical chromosomal ParA sequence (Supplementary Fig. 5b).

Since ParB did not control ParA localization in *B. burgdorferi*, we hypothesized that another factor fulfilled this function. Interestingly, in *B. burgdorferi*, the *parA* and *parB* genes are located on the chromosome in opposite, head-to-head orientations (Fig. 5a), instead of being joined in a two-gene operon as in many other bacteria[61]. This opposing gene organization is unique to Borreliaceae as *parA* and *parB* are organized in an operon structure in other spirochetes

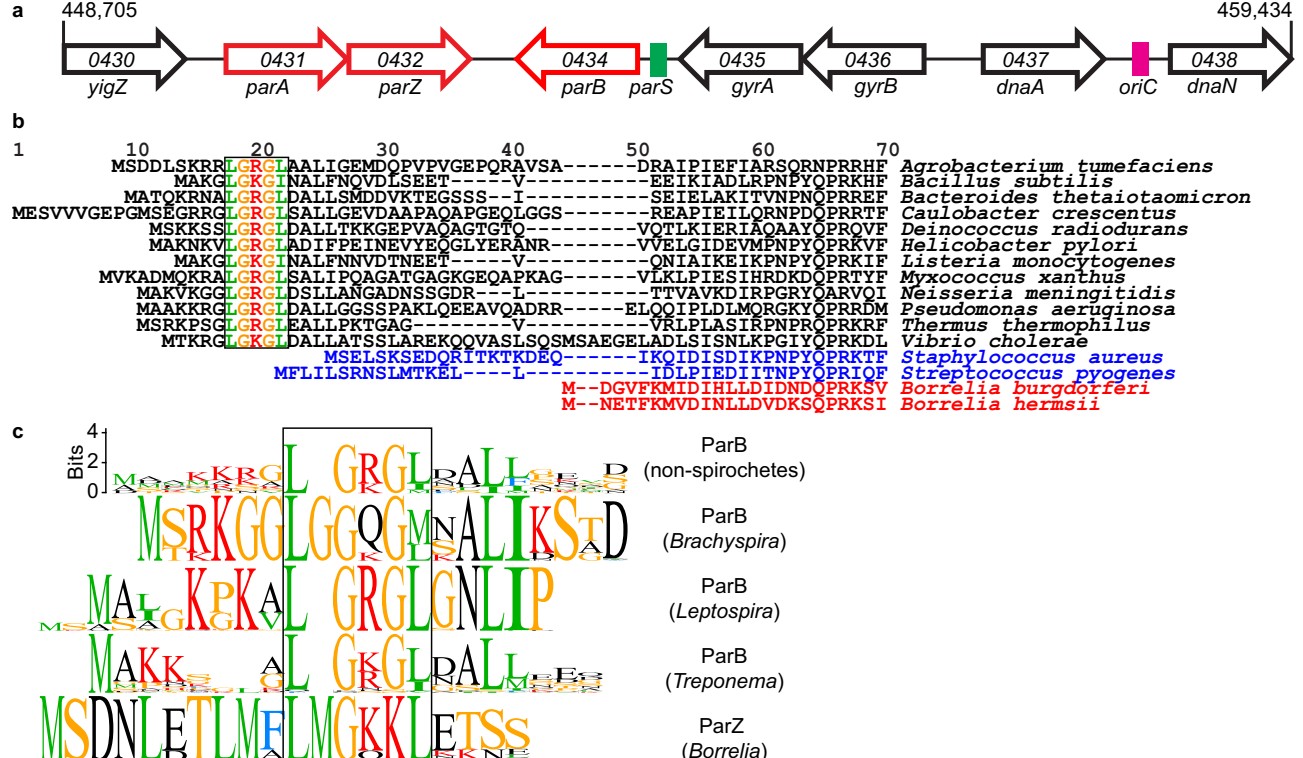

**Fig. 5 | *B. burgdorferi* ParB lacks the N-terminal peptide that controls ParA localization. a** Schematic of the *oriC*-proximal chromosome region of *B. burgdorferi* strain B31, between nucleotides 448,705 and 459,434, showing the *par* genes. Features are not drawn to scale. We propose renaming the gene *bbO432* of hypothetical function to *parZ*. **b** Alignment of chromosomally encoded ParB sequences from the indicated species. The first 70 positions of the alignment are shown here, while the full alignment is displayed in Supplementary Fig. 5a. Highlighted is the conserved LG(R/K)GL motif of ParB's N-terminal ParA ATPase-stimulating peptide, which is absent from *Borrelia*, *Streptococcus*, and *Staphylococcus* ParB sequences. **c** Sequence logos for the N-terminal 20 amino acids obtained by ClustalW alignment of unique *Borrelia* ParZ sequences as well as unique ParB sequences from selected non-spirochete bacteria (see panel **b**) or from spirochete *Leptospira*, *Brachyspira*, or *Treponema* genera. Only chromosome I sequences were considered for *Leptospira*. Boxed in are motifs similar to the conserved motif found within the N-terminal peptide of the more diverse chromosomal ParB sequences shown in (**b**).

(Supplementary Fig. 5c). Furthermore, *parA* appears to form an operon with *bbO432*, a gene of hypothetical function that we propose to rename *parZ* (Fig. 5a). We identified putative *parAZ* operons among the 29 sequenced Lyme disease *Borrelia* strains that we examined (Supplementary Fig. 5d). ParZ proteins are also well conserved across the entire sequence among the larger Borreliaceae family, including relapsing fever spirochetes (Supplementary Fig. 5e), but are not encoded by other spirochete genomes. Importantly, deletion of *parZ* in *B. burgdorferi* drastically altered the distribution of the ParA-msfGFP signal (Fig. 7e): the banded pattern of ParA-msfGFP disappeared and the fluorescent signal became more distributed within the cell, forming only patches, likely due to the known cooperativity of ParA binding to the DNA[37,62]. Additionally, in the Δ*parZ* background, we did not detect depletion of the ParA-msfGFP signal from the vicinity of *oriC*, nor did we see banded accumulation of ParA-msfGFP in cellular regions located between the *oriC* loci (Fig. 7f, g). We note that the ParA-msfGFP expression level was slightly increased by ~25% in the absence of ParZ (Supplementary Fig. 6a). However, this small difference did not cause the loss of the ParA-msfGFP banded pattern, as the ParA-msfGFP banded pattern still required ParZ even when the fusion was overexpressed by ten-fold from a shuttle vector (Supplementary Fig. 6b, c). Thus, ParZ regulates the subcellular localization of ParA in *B. burgdorferi*.

In many bacteria, ParB controls ParA activity via a basic catalytic residue[38] that lies within a conserved motif, LG(R/K)GL, located in the N-terminal peptide (Fig. 5b). Related motifs can also be found within the N-terminal peptides of chromosomal ParB sequences from spirochetes outside Borreliaceae (Fig. 5c). Intriguingly, ParZ sequences

had a similar motif in their well-conserved N-terminal peptide (Fig. 5c and Supplementary Fig 5e). Removal of this N-terminal ParZ peptide (ParZΔN20) was sufficient to disrupt ParA-msfGFP localization (Fig. 7h), even though the peptide deletion had no apparent effect on transcription (Supplementary Fig. 6d). Collectively, our data indicate that ParZ substitutes ParB's function in controlling ParA localization using a similar N-terminal motif.

## ParZ is a novel bacterial centromere-binding protein

If ParZ substitutes ParB in ParA-mediated chromosome segregation, we reasoned that ParZ must directly or indirectly bind DNA close to *oriC* to explain the ParA depletion at mCherry-ParB foci (Fig. 7b, c). Indeed, chromosomal replacement of *parZ* with *parZ-msfgfp* revealed regularly spaced fluorescent foci (Fig. 8a, WT) that resembled *oriC* labeling by mCherry-ParB (Figs. 1a, 8b). The copy number and density of the ParZ-msfGFP foci were similar to those of the mCherry-ParB foci (Supplementary Fig. 6e, compare the two control strains) and the ParZ-msfGFP and mCherry-ParB foci colocalized (Fig. 8c, d). As ParZ-msfGFP formed foci in the absence of *parBS* or *parA* (Fig. 8a), the ParZ localization profile suggested the presence of a novel centromere-like region near *oriC*. Indeed, ChIP-seq experiments using ParZ-msfGFP identified a specific enrichment region that included the *parAZ* region (Fig. 6). Strikingly, this enrichment peak, which spread over a total of ~8 kilobases of DNA sequence, was adjacent to but distinct from the mCherry-ParB ChIP-seq peak (Fig. 6, Supplementary Fig. 4a, b). Importantly, the ParZ-msfGFP ChIP-seq peak was preserved in the Δ*parBS* background, as was the mCherry-ParB peak in the Δ*parAZ* background (Fig. 6), in agreement with our microscopy observations

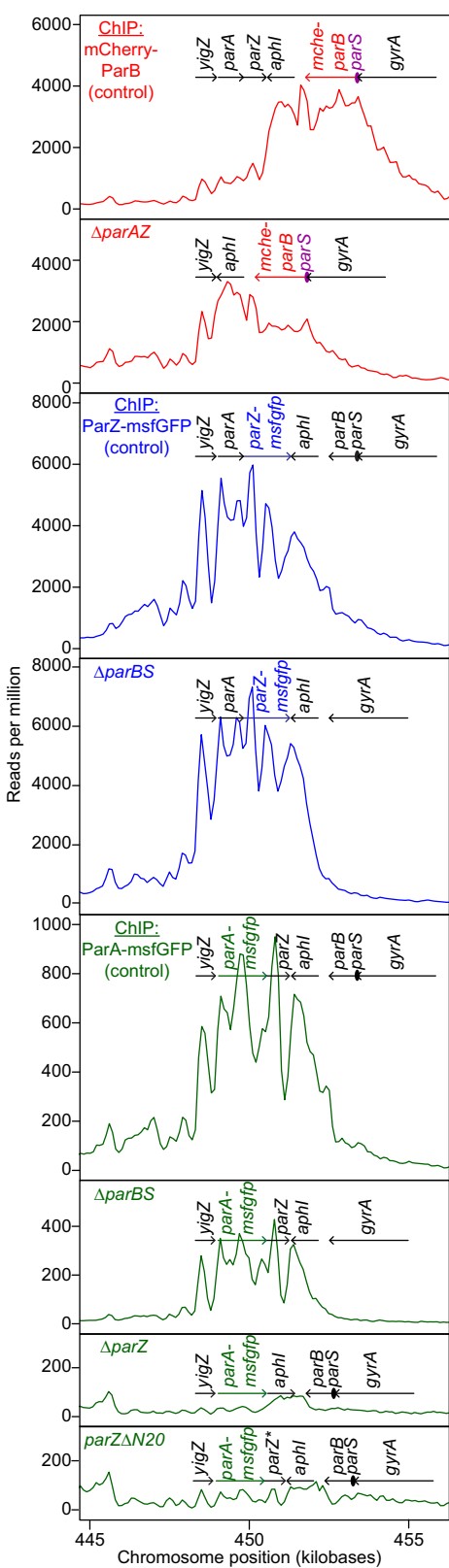

**Fig. 6 | *B. burgdorferi* ParB, ParZ, and ParA bind to the *par* locus.** ChIP-seq profiles showing the binding of mCherry-ParB (red traces), ParZ-msfGFP (blue traces), or ParA-msfGFP (green traces) to the *par* region of the *B. burgdorferi* chromosome in the indicated strain backgrounds. Strains used, from top to bottom, are: CJW_Bb379, CJW_Bb525, CJW_Bb378, CJW_Bb524, CJW_Bb488, CJW_Bb520, CJW_Bb519, and CJW_Bb610. Genes underlining the enriched sequence reads are highlighted. *aphI*, kanamycin resistance gene introduced during strain generation.

(Fig. 8a, b). If ParZ binds within its own gene, it predicts that ectopic expression of *parZ* on a shuttle vector will increase the number of fluorescent ParZ-msfGFP puncta given that the shuttle vector is in ~5-fold higher copy number than the chromosome[22,23,63]. This is indeed what we observed (Supplementary Fig. 4f, g). Additionally, the presence of an empty shuttle vector in the strain expressing ParZ-msfGFP from the endogenous locus did not cause an increase in the number of ParZ-msfGFP puncta (Supplementary Fig. 4g). Furthermore, ChIP-seq in strain CJW_Bb101 carrying *parZ-msfgfp* on the shuttle vector confirmed binding of ParZ within the shuttle vector sequence (Supplementary Fig. 4c, d), in addition to the native *parZ* region on the chromosome (Supplementary Fig. 4e). Altogether, these observations suggest that ParZ binds within its own gene.

In traditional ParA/ParB systems, ParB bound to *parS* and adjacent sequences controls ParA localization by transiently interacting with ParA[33]. If ParZ works via a similar mechanism in *B. burgdorferi*, it predicts a transient interaction between ParZ and ParA. ChIP-seq experiments verified this prediction as the ParA-msfGFP enrichment peak was at the same location as the ParZ-msfGFP peak, albeit at a lower level consistent with a transient interaction (Fig. 6 and Supplementary Fig. 4a). We also observed low-level, non-uniform mapping of ParA-msfGFP ChIP-seq reads to *B. burgdorferi*'s endogenous plasmids (Supplementary Fig. 4a, h). Since free GFP, ParZ-msfGFP, ParA-msfGFP, and mCherry-ParB pulldowns generated almost identical traces in this region (Supplementary Fig. 4h), we believe these traces represent non-specific landscape binding. ParA-msfGFP binding to the *par* locus, while reduced in the *parBS* mutant, was nevertheless still present (Fig. 6), indicating that ParA recruitment to the *par* locus did not require *parBS*. In contrast, ParA-msfGFP recruitment to the *par* locus required ParZ and, more specifically, its N-terminal peptide (Fig. 6). These results indicate that ParZ is a newly identified centromeric protein, which, similarly to ParB in other bacteria[38,39], uses its N-terminal peptide to regulate ParA localization.

**ParA, ParB, and ParZ jointly control chromosomal *oriC* spacing**
Given that ParZ, and not ParB, controls ParA in *B. burgdorferi*, we investigated the role of each Par protein in *oriC* segregation using a comprehensive set of *par* gene deletion mutants. In the absence of an appropriate setup for live-cell timelapse imaging of *B. burgdorferi*, we analyzed static snapshots of exponentially growing populations of cells. Visual inspection of the control strains that express mCherry-ParB or ParZ-msfGFP to label *oriC* revealed a near-equidistant spacing of *oriC* copies along the cell length (Fig. 8a, b, WT). For quantification, we calculated how much the *oriC* distribution in each cell of a population deviates from uniform spacing (see methods). This metric, referred to as deviation from uniform spacing (or DUS), gives a value of 0 when a cell displays a perfectly equidistant distribution of *oriC* copies. In contrast, a random intracellular distribution of *oriC* copies yields an average DUS value of 0.7 (Fig. 8e and see methods). Control strains, which have a complete set of *parA, parZ,* and *parB* genes, had very similar DUS distributions centered around ~0.2 (Fig. 8e), indicating near-equidistant spacing. Deletion of *parBS* reduced the number of *oriC* foci (Supplementary Fig. 6e) but had little impact on the regularity of *oriC* spacing (Fig. 8a). This was reflected in the DUS distribution, which displayed only a slight shift towards larger values relative to the control strains (Fig. 8f). In contrast, deletion of *parA* or *parAZ* resulted in some disruption of *oriC* spacing (Fig. 8a, b). This was quantitatively reflected in larger, rightward shifts of DUS distributions (Fig. 8f, g). Surprisingly, deletion of *parZ* alone caused a larger disruption of *oriC* spacing than the removal of *parA* or *parAZ* (Fig. 8g). It may suggest that a dysregulated ParA activity is more detrimental than no ParA activity at all. The combination of Δ*parA* and Δ*parBS* in strain CJW_Bb616 caused the strongest *oriC* spacing defect. This defect manifested as multiple *oriC* foci clustered in one or several short cell segments flanked by large cellular spaces containing DNA devoid of *oriC* spots

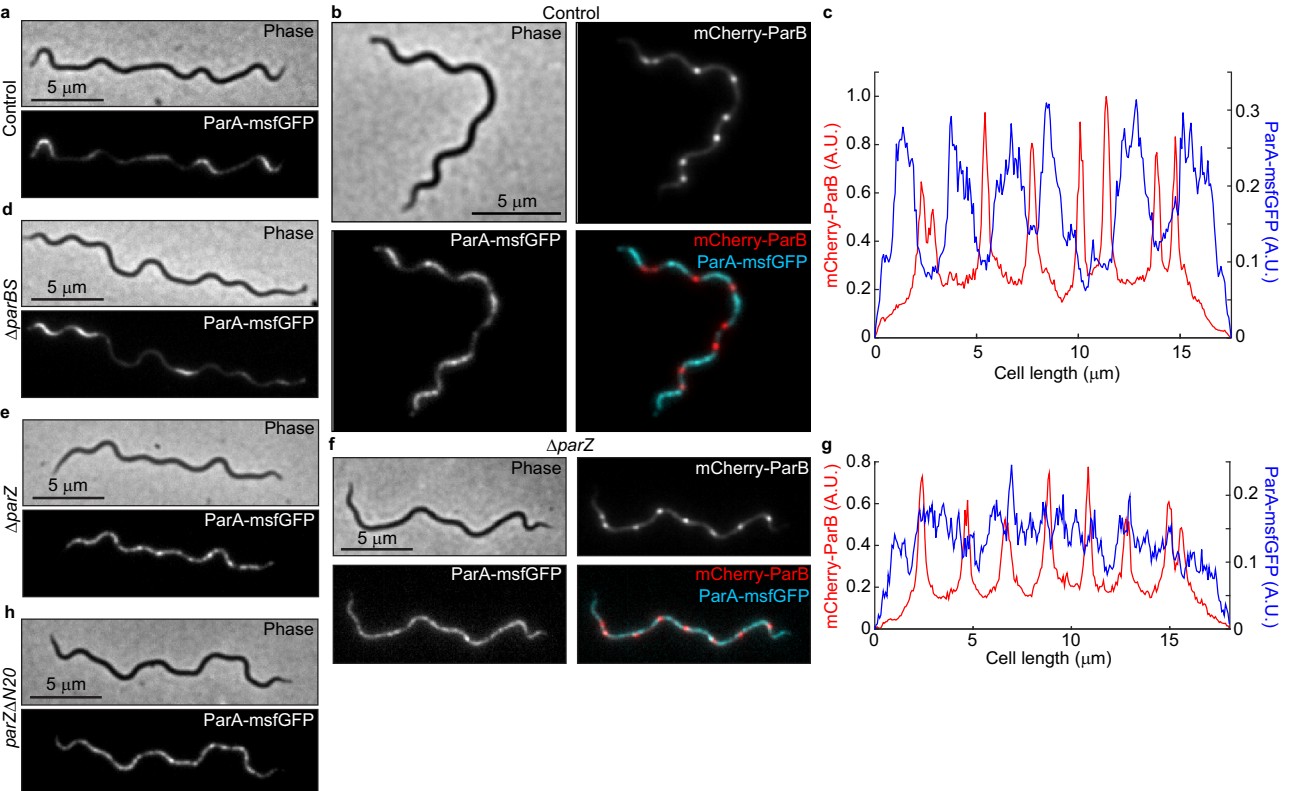

**Fig. 7 | ParZ, not ParB, controls ParA localization in *B. burgdorferi*. a** Images of ParA-msfGFP localization in knock-in strain CJW_Bb488, which carries a full complement of *par* genes and thus serves as control for ParA-msfGFP localization. **b** Images of a cell of strain CJW_Bb538 expressing ParA-msfGFP and in which *oriC* copies were localized by expression of mCherry-ParB. **c** Fluorescence intensity profiles along the cell length for the cell shown in (**b**). **d** Images of ParA-msfGFP localization in knock-in strain CJW_Bb520 in which *parBS* is deleted. **e** Images of

ParA-msfGFP localization in knock-in strain CJW_Bb519 in which *parZ* is deleted. **f** Images of a cell of strain CJW_Bb619 expressing ParA-msfGFP and in which *parZ* is deleted and *oriC* copies are localized by expression of mCherry-ParB. **g** Fluorescence intensity profiles along the cell length for the cell shown in (**f**). **h** Images of ParA-msfGFP localization in knock-in strain CJW_Bb610 in which *parZ* is missing the sequence encoding the N-terminal 20 amino acids (*parZΔN20*).

(Fig. 8a). This striking phenotype resulted in the largest DUS distribution shift towards higher values (Fig. 8f, g). Furthermore, the severe *oriC* segregation defect seen in the Δ*parA*Δ*parBS* double mutant was accompanied by a large increase in the frequency of cells that lacked *oriC* foci, from <0.3% in control strains to 3.5% in the Δ*parA*Δ*parBS* double mutant (Supplementary Fig. 6e). Collectively, our findings indicate that *parA/parZ* and *parB/parS* jointly control *oriC* segregation in *B. burgdorferi*, with *parA/parZ* playing a more prominent role.

### ParB recruits SMC to the chromosomal replication origins

Despite having lost its ability to control ParA, *B. burgdorferi* ParB still participates in *oriC* segregation (Fig. 8f). Therefore, we investigated whether ParB achieves this function via interaction with another protein. In other bacteria, ParB recruits structural maintenance of the chromosome (SMC) complexes to the *oriC* region[64–66]. The recruitment of SMC to *oriC* and subsequent organization of the *oriC* region have been shown to facilitate *oriC* segregation in *Caulobacter crescentus* and *Bacillus subtilis*[64–70]. While lacking the ParA control motif, the *B. burgdorferi* ParB protein has retained a predicted SMC-binding region located within the N-terminal domain of ParB[66] (Supplementary Fig. 5a). *B. burgdorferi* also encodes a homolog of SMC at locus *bb0045*. We therefore replaced *smc* with an *mcherry-smc* fusion both in the wild-type background and in a strain that expresses ParZ-msfGFP to visualize the *oriC* region. mCherry-SMC had no detectable effect on *oriC* copy number (Supplementary Fig. 6e) and formed fluorescent foci that colocalized with some, though not all, ParZ-msfGFP-decorated *oriC* loci (Fig. 9a, b). Consistent with this colocalization, ChIP-seq

experiments revealed enrichment of mCherry-SMC at the *par* locus, centered around the *parS* site, which is consistent with its recruitment by ParB (Fig. 9c). Deletion of *parBS* abrogated the formation of clear mCherry-SMC puncta (Fig. 9a, b), resulting in a loss of colocalization between mCherry-SMC and ParZ-msfGFP-decorated *oriC* loci. Finally, the Δ*smc* mutation caused a similar reduction in *oriC* copy number and density as Δ*parBS* (Supplementary Fig. 6e) and mildly disrupted *oriC* spacing (Fig. 9d, e). These results support the idea that ParB recruits SMC near *oriC*, and that this interaction plays a supporting role in *oriC* segregation in *B. burgdorferi*. The synergy between the ParA/ParZ and ParB/SMC systems in *oriC* partitioning was substantiated by the presence of a synthetic growth defect when both systems were defective (Δ*parAZBS* background), as measured both in semisolid plates and liquid cultures (Supplementary Fig. 6g, h).

## Discussion

Our live-cell imaging provides definite proof that the *B. burgdorferi* chromosome and plasmids exist in multiple copies during the growth phase of this spirochete (Figs. 1, 2, Supplementary Figs. 1c, 2). This dispels the notion that Lyme disease agents such as *B. burgdorferi* differ in ploidy from the relapsing fever agent *Borrelia hermsii*, which was shown to have multiple genome copies per cell using biochemical population-level measurements[24,71]. We note that fragmentation of DNA staining had been reported in air-dried *B. hermsii* and *B. burgdorferi* cells[24,72]. However, comparison with biochemical quantification of genome copy numbers[24] demonstrate that this approach is probably not reliable to assess ploidy. This is not surprising given that cellular desiccation is well known to result in double-stranded DNA

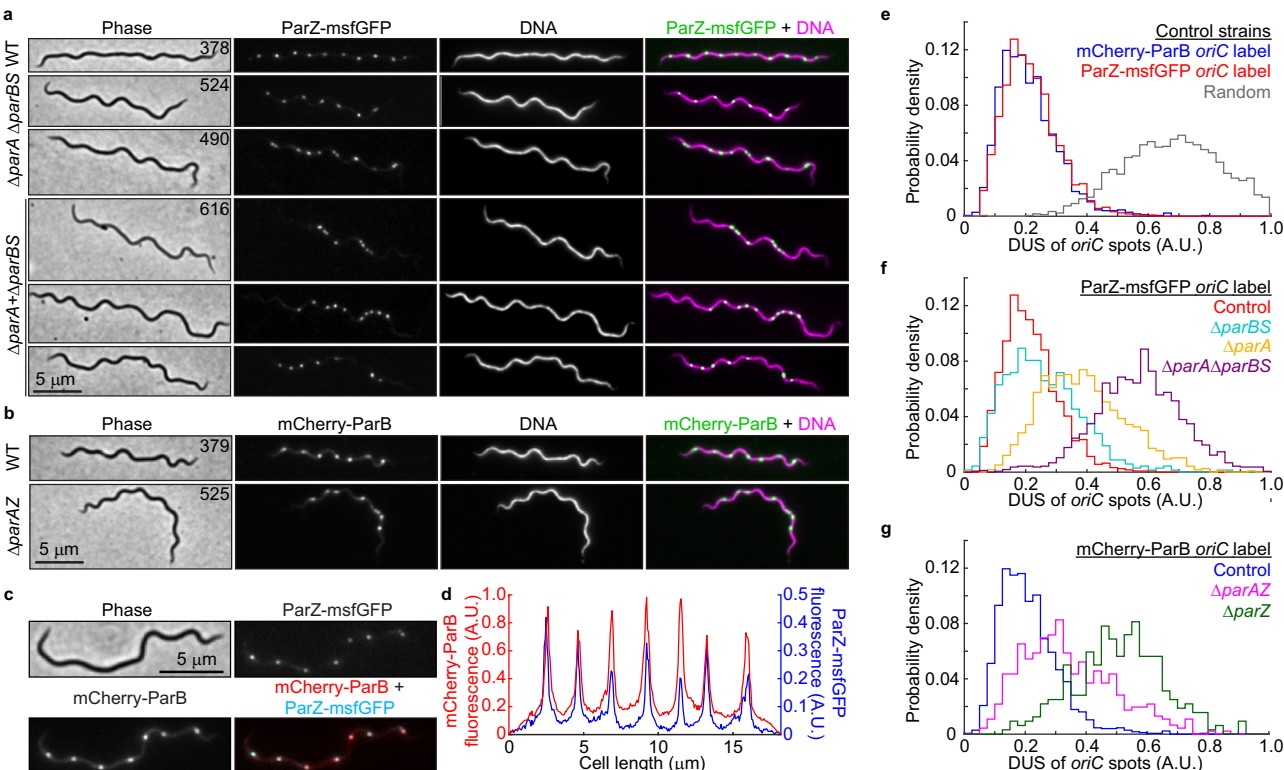

**Fig. 8 | ParZ is a novel bacterial centromere-binding protein that contributes to *oriC* segregation. a** Images of ParZ-msfGFP signal in strains carrying the full complement of *B. burgdorferi par* genes (WT) or lacking the *par* genes and/or *parS* sequence as indicated. DNA was stained with Hoechst 33342. CJW_Bb strain numbers are listed on the phase-contrast images. **b** Images of mCherry-ParB signal in strains carrying the full complement of *par* genes (WT) or lacking the *parAZ* operon, as indicated. DNA was stained with Hoechst 33342. CJW_Bb strain numbers are listed on the phase contrast images. **c** Images of strain CJW_Bb403 showing colocalization of ParZ-msfGFP and mCherry-ParB signals. **d** Fluorescence intensity profile along the cell length for the cell shown in (**c**). **e** Histograms showing the distributions of deviations from uniform spacing (DUS, see methods) of *oriC* spots

in control strains expressing mCherry-ParB (strain CJW_Bb379) or ParZ-msfGFP (strain CJW_Bb378) to label *oriC*. Also shown in gray are DUS values obtained by simulating a random redistribution of *oriC* loci in the analyzed cells of strain CJW_Bb379 (see methods). **f** Histograms comparing DUS distributions of ParZ-msfGFP-labeled *oriC* foci in control (CJW_Bb378), Δ*parBS* (CJW_Bb524), Δ*parA* (CJW_Bb490), and Δ*parA* Δ*parBS* (CJW_Bb616) strains. **g** Histograms comparing DUS distributions of mCherry-ParB-labeled *oriC* foci in control (CJW_Bb379), Δ*parAZ* (CJW_Bb525), and Δ*parZ* (CJW_Bb626) strains. Source data for panels (e–g) are provided as a Source Data file. The numbers (*n*) of cells analyzed and the number of replicates are provided in Supplementary Data 2.

breaks. In contrast, our imaging results using live *B. burgdorferi* cells are in good agreement with qPCR analysis (Fig. 1f-g and Supplementary Fig. 1h). Interestingly, qPCR experiments have also demonstrated the polyploid nature of the syphilis spirochete *Treponema pallidum*[73]. Thus, polyploidy may reflect a general property of spirochetes.

Polyploidy may be advantageous under stress. For instance, the presence of multiple genome copies has been shown to increase resistance to DNA damage in cyanobacteria[74], presumably by facilitating DNA recombination and repair. Polyploid cyanobacteria can also recover from infection by lytic phages as long as not all genome copies are degraded prior to phage inactivation[75]. In the context of a pathogen like *B. burgdorferi*, polyploidy may facilitate persistence in the vertebrate host. Modeling studies have predicted that antigenic variation systems facilitate longer survival of pathogens in immunocompetent hosts when they are present in multiple genomic copies than when encoded on a single-copy genome[71]. The plasmid lp28-1 contains an antigen variation system[76]. Its presence in multiple copies (Fig. 2) may further help this spirochete evade the host antibody response. *B. burgdorferi* polyploidy also has practical implications, one being *B. burgdorferi* load assessment in animal tissues by qPCR which often assumes that one genome copy represents one bacterial cell[19]. Actual tissue bacterial loads could be ~10-fold lower.

In this study, we also demonstrated the regular subcellular distribution of both chromosome and plasmid copies (Figs. 1a, 2a, 8e and Supplementary Fig. 1c). In the context of symmetric division (i.e., at

midcell)[28], such a regular linear distribution of a multicopy genome ensures that each daughter cell inherits near-equal copies of the genome, as in cyanobacteria[52,53]. Indeed, severe disruption of near-uniform distribution of the *B. burgdorferi* chromosome in the Δ*parA*Δ*parBS* mutant (Fig. 8a,f) was accompanied by chromosome inheritance defects (Supplementary Fig. 6e). Regular genome patterning also implies the existence of active segregation mechanisms. For the *B. burgdorferi* plasmids, the conserved plasmid maintenance loci[31] are likely responsible for their observed subcellular distribution. For bacterial chromosomes, segregation of the *oriC* regions generally involves ParB-dependent control of ParA activity. Unexpectedly, we found that in *B. burgdorferi*, and presumably other *Borrelia* species, the localization of ParA is instead primarily controlled by a novel centromere-binding protein, which we named ParZ (Figs. 5–8, Supplementary Figs. 4–6). Control of ParA by factors other than ParB has previously been documented in *Myxococcus xanthus*, where the CTPase PadC recruits ParA to subpolar bactofilin structures[35,77], although *M. xanthus* ParB retains its ParA-stimulating peptide (Fig. 5b) and the ability to control ParA localization[78]. In *B. burgdorferi*, we envision that the ParA/ParZ system harnesses forces within the cell to drive segregation of the *oriC* loci, similarly to how the ParA/ParB system drives patterning of multicopy plasmids[33,42,49].

ParZ displays several other properties reminiscent of ParB. Like ParB, ParZ appears to spread on the DNA around its cognate centromeric site, occupying several kilobases of DNA sequence (Fig. 6).

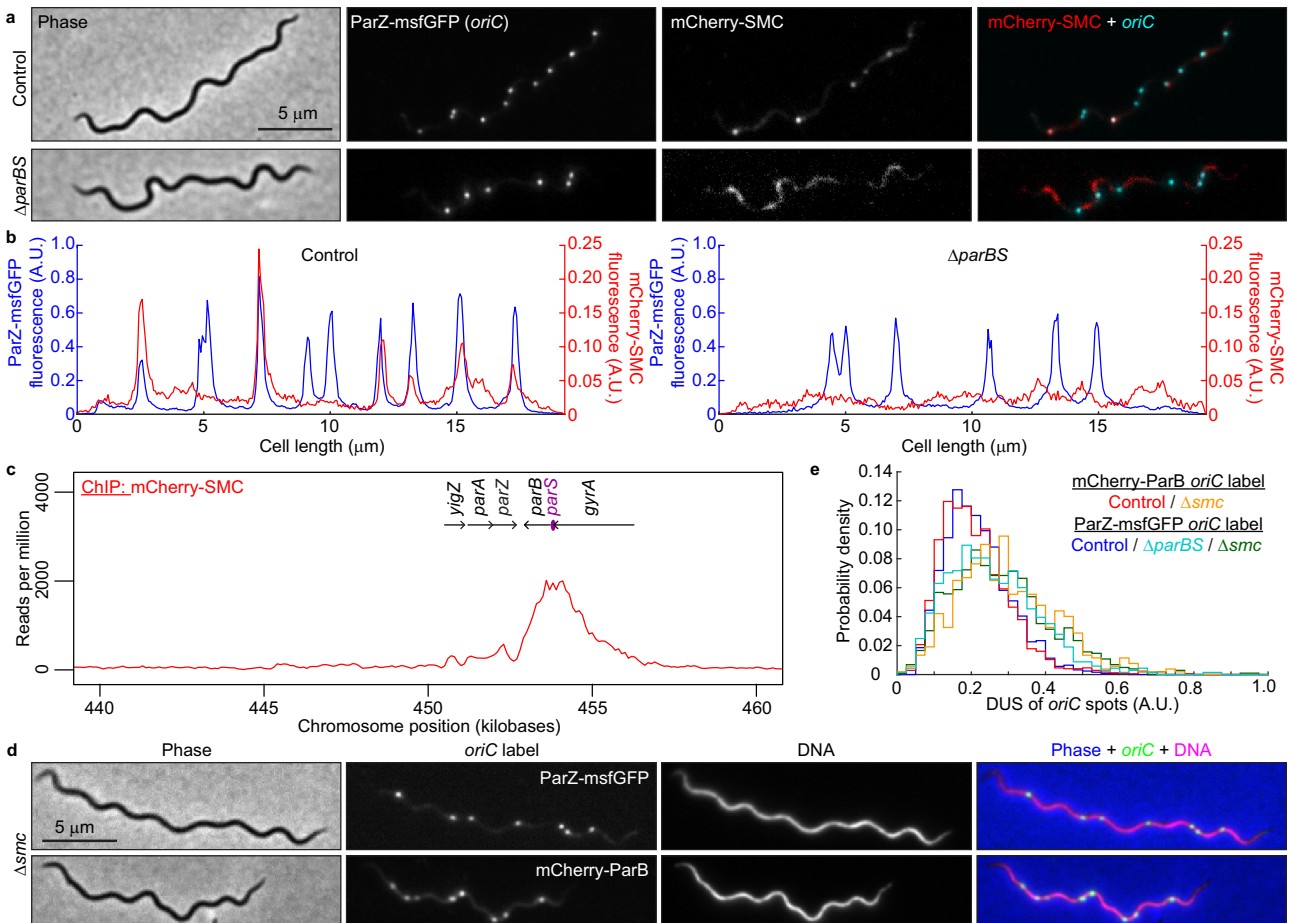

**Fig. 9 | SMC is recruited to the *oriC* region by ParB and mildly contributes to regular *oriC* spacing. a** Images of cells expressing ParZ-msfGFP (to label *oriC*) and mCherry-SMC in control (CJW_Bb602, carrying the full complement of *par* genes) and Δ*parBS* (CJW_Bb615) strains. **b** Fluorescence intensity profiles along the cell length for the cells shown in (**a**). **c** ChIP-seq profile showing the binding of mCherry-SMC to the *par* locus in strain CJW_Bb601. Genes underlining the enriched sequence reads are highlighted. **d** Images of cells showing *oriC* localization in Δ*smc* strains in which the *oriC* is labeled with either ParZ-msfGFP (CJW_Bb603) or mCherry-ParB (CJW_Bb604). Hoechst 33342 was used to stain the DNA. An overlay is also shown. **e** Histograms showing DUS of *oriC* spots in cells of control and Δ*smc* strains in which *oriC* was labeled with either ParZ-msfGFP (control, CJW_Bb378; Δ*smc*, CJW_Bb603) or mCherry-ParB (control, CJW_Bb379; Δ*smc*, CJW_Bb604). Also shown is the DUS histogram for the Δ*parBS* strain CJW_Bb524. Source data are provided as a Source Data file. The numbers (*n*) of cells analyzed and the number of replicates are provided in Supplementary Data 2.

Additionally, ParZ binding to the DNA is not uniformly distributed; rather, it displays several peaks and valleys, which was also observed in the case of ParB[79,80]. This broad distribution of ParZ binding likely increases the probability of interaction with ParA. Future investigation will be required to uncover the mechanisms underlying ParZ spreading, including whether it shares any mechanistic similarities with ParB, such as CTPase (or other NTPase) activity, clamp formation, and lateral sliding along the DNA[33–35,81]. In *B. burgdorferi*, interestingly, ParB spread from the *parS* site appears skewed in one direction that matches that of the transcription of the bound genes (Fig. 6). In turn, binding of ParB to the DNA decreases when the bound genes become convergently transcribed with the direction of ParB spread (Fig. 6 and Supplementary Fig. 4b). We observed a similar behavior for ParZ in several genetic backgrounds (Fig. 6 and Supplementary Fig. 4b, d, e). This raises the possibility that active transcription promotes the spread of ParB and possibly ParZ away from the loading site when the direction of spread and transcription are the same and limits the spread when transcription and spread occur in opposite directions. A similar inference can be gleamed from ChIP-seq traces of ParB binding to the DNA in *Corynebacterium glutamicum*[82]. Additionally, transcription of underlying genes was invoked to explain asymmetric spread of ParB onto *parS*-proximal DNA sequences in *C. crescentus*[79].

While ParB lost its ability to control ParA localization in *B. burgdorferi*, it retained its role in recruiting SMC to *oriC* (Fig. 9), thereby contributing to *oriC* partitioning, though to a lesser degree than the ParA/ParZ system (Fig. 8f). SMC, possibly through the organization of the *oriC* region[69], also promotes *oriC* segregation. This is evident from the similar *oriC* segregation defects displayed by the Δ*parBS* and Δ*smc* mutants (Fig. 9e). The ParB/SMC system, as well as ParA, may also be involved in initiation of DNA replication[83], as several *par* and *smc* mutants that we characterized have decreased *oriC* copy numbers during exponential growth in culture (Supplementary Fig. 6e).

During the evolution of Borreliaceae, ParB retained a SMC loading function but remarkably transferred its ParA-regulating function to a new player. How could it have happened? BLAST searches allowed us to identify distant homologs of ParZ among the Firmicutes and Fusobacteria (Supplementary Fig. 7). The *Streptococcus*, *Staphylococcus*, *Enterococcus*, and *Bacillus* genera are particularly well represented among the Firmicutes hits (Supplementary Fig. 7). These ParZ-like proteins contain only short regions of homology with the Borreliaceae ParZ proteins and do not include the ParB-like N-terminal peptide. More intriguing though is that a subset of these ParZ-like proteins are encoded by phages (Supplementary Fig. 7). It is tantalizing to speculate acquisition of *parZ* from a phage infection of the spirochete ancestor

of the Borreliaceae. In fact, *Borrelia* genomes carry extensive evidence of infection by phages, including those related to known streptococcal phages[18]. Phage insertion into the *parB* gene may have fused the sequence encoding the N-terminal, ParA-controlling peptide of ParB to a phage-encoded protein that binds its own DNA sequence. Subsequent genetic drift of the phage-disrupted spirochete genome may have resulted in inversion of the orientation of the *parB* gene and loss of the remaining phage sequence, generating the current locus structure. While this evolutionary scenario remains speculative, our findings highlight the plasticity and evolvability of microbial genomes, even in the context of fundamental cellular functions such as genome partitioning. Understanding basic biological functions in non-model species is important, particularly in bacterial pathogens where divergent regulatory mechanisms of essential bacterial functions may provide new targets for specific therapeutic intervention.

## Methods

### Bacterial strains and growth conditions

The following *Escherichia coli* cloning strains were used to generate, maintain, and grow the various plasmids used in this study: DH5α (Promega), NEB 5-alpha and NEB 5-alpha F′*I�q* (New England Biolabs), and XL10-Gold (Agilent). The strains were grown on Luria Bertani agar plates incubated at 30 °C or in Super Broth (35 g/L bacto-tryptone, 20 g/L yeast extract, 5 g/L NaCl, and 6 mM NaOH) liquid medium with shaking at 30 °C. Antibiotics were used at the following concentrations: kanamycin at 50 µg/mL, gentamicin at 15 to 20 µg/mL, spectinomycin or streptomycin at 50 µg/mL, rifampin at 25 µg/mL in liquid culture or 50 µg/mL in plates, and hygromycin B at 100 to 200 µg/mL.

*B. burgdorferi* strains and their generation are detailed in Supplementary Data 1, Worksheet 1. Requests for the *B. burgdorferi* strains generated in this study should be directed to Christine Jacobs-Wagner. These strains were grown in complete Barbour-Stoenner-Kelly (BSK)-II liquid medium in humidified incubators at 34 °C under 5% $CO_2$ atmosphere[6,10,11]. Complete BSK-II medium contained 50 g/L bovine serum albumin (Millipore #810036), 9.7 g/L CMRL-1066 (US Biological #C5900-01), 5 g/L Neopeptone (Difco #211681), 2 g/L Yeastolate (Difco #255772), 6 g/L HEPES (Millipore #391338), 5 g/L glucose (Sigma-Aldrich #G7021), 2.2 g/L sodium bicarbonate (Sigma-Aldrich #S5761), 0.8 g/L sodium pyruvate (Sigma-Aldrich #P5280), 0.7 g/L sodium citrate (Fisher Scientific #BP327), 0.4 g/L *N*-acetylglucosamine (Sigma-Aldrich, #A3286), 60 mL/L heat-inactivated rabbit serum (Gibco #16120), and had a pH of 7.60. Complete BSK-H media was acquired from Sigma-Aldrich (#B8291). Tubes contained 6 to 7 mL or 13 to 15 mL culture depending on the tube size used (8-mL volume, Falcon, #352027, or 16-mL volume, Falcon, #352025) and were kept tightly closed. Any larger volume vessels were kept loosely capped in the incubator. BSK-1.5 medium for plating was previously described[12,84,85] and contained 69.4 g/L bovine serum albumin, 12.7 g/L CMRL-1066, 6.9 g/L Neopeptone, 3.5 g/L Yeastolate, 8.3 g/L HEPES, 6.9 g/L glucose, 6.4 g/L sodium bicarbonate, 1.1 g/L sodium pyruvate, 1.0 g/L sodium citrate, 0.6 g/L *N*-acetylglucosamine, and 40 mL/L heat-inactivated rabbit serum, and had a pH of 7.50. Selection antibiotics were used at the following final concentrations in both liquid cultures and plates: kanamycin at 200 µg/mL[86], gentamicin at 40 µg/mL[87], streptomycin at 100 µg/mL[88], hygromycin B at 300 µg/mL[85,89] and blasticidin S at 10 µg/mL[89]. Piperacillin was used at a final concentration of 10 ng/mL. Unless otherwise indicated, all cultures were maintained in exponential growth by diluting the cultures into fresh medium before reaching ~5 × $10^7$ cells/mL.

Semisolid BSK-agarose plating medium[85] contained 2 parts of 1.7% agarose solution in water and 3 parts BSK-1.5 medium containing appropriate amounts of selective antibiotics, as needed, to yield in the final plating mix the concentrations listed above. Each plate was seeded with a maximum of 1 mL *B. burgdorferi* culture. The agarose, melted and maintained at 55 °C, and the BSK-1.5 medium, briefly pre-

equilibrated at 55 °C, were mixed and then 25 mL were poured into each *B. burgdorferi*-seeded plate. The plates were swirled gently to mix, allowed to solidify at room temperature (RT) in a biosafety cabinet for ~ 30 min, then transferred to a humidified $CO_2$ incubator and incubated between 10 days and 3 weeks.

### Genetic manipulations

All plasmids (Supplementary Data 1, Worksheet 2) were generated using standard molecular biology techniques that included ligation of restriction endonuclease-digested plasmids and PCR products, Gibson assembly[90] of DpnI-digested PCR products using New England Biolabs' platform, or site-directed mutagenesis using the Agilent Quick Change Lightning Site-Directed Mutagenesis kit. Sequences of the oligonucleotide primers used in the course of plasmid generation are provided in Supplementary Data 1, Worksheet 3. The plasmids, for which relevant sequences were confirmed by Sanger DNA sequencing at Quintara Biosciences, were introduced into *E. coli* host strains by heat shock or electroporation and were archived as *E. coli* strains. Minipreps were done with Zymo Research Zippy plasmid miniprep kit. When requesting any plasmid, we urge the requestor to provide us with the *E. coli* CJW strain number listed in Supplementary Data 1, Worksheet 2, in addition to the name of the plasmid.

Large amounts of plasmid DNA were isolated from saturated 50 mL Super Broth *E. coli* cultures using the Qiagen Plasmid Plus Midi kit with the final elution performed in water. *E. coli* / *B. burgdorferi* shuttle vectors (25 to 50 µg) were electroporated into 50-100 µL volumes of *B. burgdorferi* competent cells, which were prepared as previously described[91] by various concentrations and washing steps using electroporation solution (93.1 g/L sucrose, American Bioanalytical #AB01900, 150 mL/L glycerol, American Bioanalytical #AB00751) and 10 min centrifugation steps at 4 °C and 10,000 × *g* in the FX6100 rotor of a Beckman Coulter Allegra X-14R centrifuge. Suicide vectors (50–75 µg) were linearized with the restriction endonucleases indicated in Supplementary Data 1, Worksheet 1, ethanol precipitated[92], resuspended in 25 µL water, then electroporated into 100 µL aliquots of *B. burgdorferi* competent cells. Electroporation was done at 2.5 kV, 25 µF, and 200 Ω[12,84] in a 2 mm-gap cuvette. The electroporated cells were immediately recovered in 6 mL BSK-II medium, and incubated overnight at 34 °C, after which selection in liquid culture and in semisolid BSK-agarose plates was performed. When needed, non-clonal, liquid-selected populations of transformants were plated in semisolid BSK-agarose for clone isolation. For each B31-derived clone, we confirmed the endogenous plasmid complement by multiplex PCR[93]. Plasmid complement was not determined for clones derived from strains other than B31 due to the absence of a complete, standardized PCR primer set for those strains. To confirm the correct homologous recombination of suicide vectors, total genomic DNA was isolated from the *B. burgdorferi* clones using the Qiagen DNeasy Blood & Tissue Kit, and diagnostic PCR was performed to confirm insertion or deletion within the targeted locus, as well as correct recombination of the flanking sequences. The genetic manipulations of the chromosome at the *par* and *smc* loci are depicted schematically in Supplementary Fig. 8. Insertion of *parS^P1* cassettes at the *phoU*, *uvrC*, and *lptD* loci occurred in intergenic regions between two convergently transcribed genes. The same strategy was used to insert the *parS^P1* cassette in *B. burgdorferi* plasmids. For the cp32 plasmids, the *parS^P1* cassette was inserted into the transposon sequence in transposon mutant backgrounds for which the transposon was mapped to have inserted in an intergenic region flanked by convergently transcribed genes[30]. As a result, the likelihood of coding sequence or promoter disruption by the genetic changes was minimized. An exception is lp25, where the cassette was inserted in the scar generated upon deletion of the nonessential gene *bbe02*.

## Mouse-tick transmission studies

**Ethics statement.** All animal work was performed according to the guidelines of the National Institutes of Health, Public Health Service Policy on Humane Care and Use of Laboratory Animals and the United States Institute of Laboratory Animal Resources, National Research Council, Guide for the Care and Use of Laboratory Animals. Protocols were approved by the Rocky Mountain Laboratories, NIAID, NIH Animal Care and Use Committee. The Rocky Mountain Laboratories are accredited by the International Association for Assessment and Accreditation of Laboratory Animal Care (AAALAC). All efforts to minimize animal suffering were made.

**Experimental mouse-tick infection studies.** Mouse (*Mus musculus*) infections were conducted with 4-8 weeks-old female RML mice, an outbred strain of Swiss-Webster mice reared at the Rocky Mountain Laboratories breeding facility. Mice were housed at an ambient temperature between 20.6 and 23.9 °C, ambient humidity of 50%±10%, and under a 12 h ON / 12 h OFF light cycle. Five mice per strain were inoculated intraperitoneally ($4 \times 10^4$ spirochetes) and subcutaneously ($1 \times 10^4$ spirochetes), with the number of injected spirochetes predetermined by Petroff-Hausser counting. Mouse infection was confirmed 3 weeks post-injection by isolation of *B. burgdorferi* from ear biopsies in BSK-II medium containing appropriate antibiotics.

Larval *Ixodes scapularis* were purchased from Oklahoma State University. *I. scapularis* were maintained between feeds at ambient light and temperature in bell jars over potassium sulfate-saturated water. Approximately 100 naïve *I. scapularis* larvae were fed to repletion per infected mouse. Acquisition and retention of *B. burgdorferi* by larval ticks was assessed 1 week after drop-off, and spirochete load was determined through mechanical disruption and plating. Two naïve mice per strain were fed upon by 15-20 infected *I. scapularis* nymphs. The number of *B. burgdorferi* in nymphs was assessed prior to feeding and 10 days after drop-off through mechanical disruption and plating. Mouse infection was confirmed 5 weeks post-nymphal tick feeding by isolation of *B. burgdorferi* from ear biopsies in BSK-II medium containing appropriate antibiotics.

## Tick midgut cryopreservation

Tick midguts were dissected and fixed on ice using 4% formaldehyde solution in phosphate buffered saline (PBS), washed thrice for 5 min with cold PBS, infiltrated with 15% sucrose in PBS at 4 °C, then with 30% sucrose in PBS at 4 °C, then with a 1:1 mix of 30% sucrose in PBS and Optimal Cutting Temperature (OCT) compound (Tissue-Tek), and finally with pure OCT compound. The tissue was then frozen in OCT compound using liquid-nitrogen-cooled isopentane, then stored at −80 °C and shipped on dry ice. Thin, 10-µm sections were cut using a cryostat.

## Microscopy

*B. burgdorferi* culture density was measured by dilution of a culture in a C-Chip disposable hemocytometer (INCYTO) and direct counting of the cells under darkfield illumination obtained using a Nikon Eclipse E600 microscope equipped with a 40×0.55 numerical aperture (NA) Ph2 phase-contrast air objective and darkfield condenser optics. For fluorescence microscopy imaging, BSK-II-grown *B. burgdorferi* strains were spotted onto a 2% agarose-PBS pad[28,94], covered with a no. 1.5 coverslip, then imaged using Nikon Eclipse Ti microscopes equipped with 100X Plan Apo 1.45 NA phase-contrast oil objectives, Hamamatsu Orca-Flash4.0 V2 CMOS cameras, and either a Sola LE light source or a Spectra X Light engine (Lumencor). The microscopes were controlled by Nikon Elements software. The following Chroma filter cubes were used to acquire the fluorescence images: DAPI, excitation ET395/25x, dichroic T425lpxr, emission ET460/50 m; GFP: excitation ET470/40x, dichroic T495lpxr, emission ET525/50 m; mCherry/TexasRed, excitation ET560/40x, dichroic T585lpxr, emission ET630/75 m. DNA

staining was obtained by incubating the culture for 15 min with Hoechst 33342 (Molecular Probes) at a final concentration of 1 µg/mL.

Tick midgut sections (see above) were processed as follows. The slides supporting the sections were brought up to RT, washed thrice with PBS, permeabilized for 30 min at RT using 0.2% Triton X-100 in PBS, blocked for 30 min at RT using 5% bovine serum albumin (BSA) (w/v) in PBS, 0.1% Tween-20, stained with RFP-Booster ATTO594 (ChromoTek) diluted 1:200 in blocking buffer (see above), washed thrice with PBS for 5 min each, and stained with Hoechst 33342 1:1000 in PBS for 10 min. The stained sections were mounted in PBS under a No 1.5 coverslip and were imaged on an OMX BLAZE microscope system (GE). Images in the DAPI, GFP, and mCherry channels were acquired at 100 nm z intervals. The image stacks were then deconvolved and registered using the Softworks software.

## Image analysis

Cell outlines were generated using phase-contrast images and the Oufti analysis software package[95], with the following parameters: Edgemode, LOG; Dilate, 2; openNum, 3; InvertImage, 0; ThreshFactorM, 0.985; ThreshMinLevel, 0; EdgeSigmaL, 1; LogThresh, 0. Cell outlines were curated as follows: (i) outlines of cell debris were manually removed; (ii) outlines of cells that curled on themselves, crossed other cells, or were partially outside the field of view were also manually removed; (iii) partial outlines of cells were manually extended when feasible; (iv) outlines were manually added in some cases where automated outline generation failed; and (v) outlines were manually split for cells displaying clear phase profile dips around midcell, which indicated that the cytoplasmic cylinders of the daughter cells had separated yet remained linked by an outer membrane bridge[28]. This was confirmed by visual inspection of the fluorescence signal(s), as phase profile midcell dips often corresponded to a dip in the fluorescence signal. For this reason, throughout the manuscript, the term cell refers to an individual cytoplasmic cylinder. Lastly, the "Refine All" function of the Oufti software was ran, a final curation removed improper outlines, and the fluorescence signal was added to the cell outline file in Oufti. For signal quantification and intensity profile generation, we used the Oufti background fluorescence subtraction and intensity profile generation features. Image visualization was done using FiJi software[96], GraphPad Prim 9.3.1 software, and Adobe Illustrator 2023.

To detect fluorescent spots, the Modified_Find_Irregular_Spots.m MATLAB function[97] was used, with the following parameters: fitRadius: 5, edgeDist: 2.5, centerDist: 1; peakRadius: 3; shellThickness: 1; quantileThreshold: 0.3. The value of the intensityRatioThreshold parameter was determined individually for each microscopy experiment and fluorescence channel. For each intensityRatioThreshold value empirically tested, visual inspection of the accuracy of spot detection was performed on a subset of the cells using the VisualizeSpotDetection.m function. Once an appropriate intensityRatioThreshold value was identified, all the cell outlines were visually inspected and the ones that displayed under- or over-detection of spots were manually removed, yielding a curated cell list. Then, spots were identified and added to this final cell list using the add_spots_to_cellList.m routine, while the data was exported into a table format using export_to_table.m and extract_field.m routines. A summary of the imaging experiments is provided in Supplementary Data 2, Worksheet 1. Each experimental sample was given a unique identifier. Thus, replicates of a given strain will have different identifiers. This summary includes the following information for each experiment: strain name, replicate number, relevant treatments, culture density at the time of image acquisition, number of cells, and intensityRatioThreshold values used to detect the fluorescent spots. The *oriC:terC* ratio was calculated from microscopy data as follows: (i) the *oriC* and *terC* copy number per cell was plotted as a function of

cell length in GraphPad Prism; (ii) a linear fit of the data was performed, and the slope of each fit was extracted; and (iii) the slope of the *oriC* fit was divided by the slope of the *terC* fit.

To quantify the extent to which the distribution of the *oriC* spots along the length of the cell deviates from an equidistant distribution, we performed the following analysis steps for each cell. First, we measured all distances between adjacent *oriC* spots, or DM (for distance measured). A cell with $n$ spots will generate $n−1$ DM values. Next, in silico and for each cell individually, we equidistantly redistributed the *oriC* spots within the length of that cell, using the calculate_distance_ratio.m routine. This analysis assumes that each *oriC* spot is separated from the adjacent *oriC* spot or from the adjacent cell end (for the first and last *oriC* spot in a given cell, respectively) by the same distance. In this scenario, the distance DE between adjacent, equidistantly redistributed spots for a given cell is:

$$DE = \frac{L}{n+1} \qquad (1)$$

where $L$ is the length of the cell and $n$ is the number of *oriC* spots in the cell. Then, for each pair of adjacent *oriC* spots, we calculated a distance ratio, DR, defined as the ratio of the measured distance between two adjacent spots and the value this distance would have if all the spots in the cell were equidistantly spaced.

$$DR_i = \frac{DM_i}{DE} \qquad (2)$$

Therefore, for a perfectly equidistant distribution of *oriC* spots, all distance ratios are exactly 1. Finally, to assess how the *oriC* spots of a given cell deviate from equidistant spacing, we calculated a deviation from uniform spacing (or DUS) metric defined as the mean per cell of the absolute values of the deviation of the DR values from 1:

$$DUS = \frac{\sum_{i=1}^{n-1} |DR_i - 1|}{n-1} \qquad (3)$$

Please note that in a cell with $n$ spots, there will be $n−1$ distances between adjacent spots. The data shown in Figs. 8e–g and 6e show the distribution of DUS values within the population of cells of the indicated strains.

We also simulated a random distribution of *oriC* copies inside the cells using the simulate_distance_ratio.m routine. For each analyzed cell (see above), this routine randomly redistributes each cell's *oriC* copies along the length of that cell, then extracts the distances between adjacent randomly redistributed *oriC* spots and calculates the DR and DUS values as above.

## Sample growth for next-generation sequencing

A summary of the ChIP-seq samples analyzed and reported in this study is given in Supplementary Data 2, Worksheet 2. For each strain and replicate, we provide information on culture growth conditions, inoculation and harvesting dates, and harvesting culture densities. The samples were prepared as follows. Exponentially growing *B. burgdorferi* strains were used to inoculate 250 mL BSK-II cultures. After two or three days of growth the cultures were fixed by addition of 95 mL 37% formaldehyde (Sigma-Aldrich #F8775) followed by incubation with rocking for 30 min at RT. Formaldehyde was then quenched by addition of 18 mL of 2.5 M glycine followed by incubation with rocking for 5 min at RT. The samples were chilled on ice for 10 min, transferred to conical centrifuge tubes and pelleted at 4 °C using a 30 min, 4300 × $g$ spin in an Allegra X-14R centrifuge (Beckman Coulter) equipped with a swinging bucket SX4750 rotor. The pellet was resuspended in 30 mL ice-cold HN buffer (50 mM NaCl, 10 mM HEPES, pH 8.0)[98], then pelleted at 4 °C and 10,000 × $g$ for 10 min in a fixed angle FX6100 rotor.

The pellet was resuspended in 1.5 mL final cold HN buffer and pelleted once more at 4 °C and 10,000 × $g$ for 10 min. Finally, the pellet was resuspended in 500 µL ice-cold ChIP buffer A (12.5 mM Tris, 12.5 mM EDTA, 62.5 mM NaCl, 25% w/v sucrose, pH 8.0), frozen in a dry ice-ethanol bath, and stored at −80 °C.

## Chromatin immunoprecipitation-sequencing

The frozen *B. burgdorferi* cells were thawed on ice. One hundred microliters of cells were used to prepare DNA samples for whole genome sequencing (see below), and 400 µL were processed for chromatin immunoprecipitation-sequencing (ChIP-seq) as described previously[99]. Briefly, the fixed cells were lysed using lysozyme at 4 mg/mL final concentration. Crosslinked chromatin was sheared to an average size of 250 bp by sonication using a Qsonica Q800R2 water bath sonicator. The lysate was precleared using Protein A magnetic beads (Fisher 45-002-511) and was then incubated with 4 µL undiluted anti-GFP antibodies[100] or anti-mCherry antibodies[67] overnight at 4 ˚C. The next day, the lysate was incubated with Protein A magnetic beads for 1 h at 4 °C. After washes and elution, the immunoprecipitate was incubated at 65 °C overnight to reverse the crosslinks. The DNA was next treated with RNase A, proteinase K, extracted with phenol:chloroform:isoamyl alcohol (25:24:1), resuspended in 100 µL of buffer EB (Qiagen), and used for library preparation with the NEBNext UltraII kit (E7645). The library was sequenced using Illumina NextSeq500 at Indiana University's Center for Genomics and Bioinformatics.

## Whole genome sequencing

For whole genome sequencing (WGS), 100 µL of the frozen cells from above were pelleted, resuspended in 100 µL of TE (50 mM Tris pH 8.0, 10 mM EDTA) containing 1 µL of proteinase inhibitor (Sigma P8340) and 6 µL of Ready-Lyse lysozyme (Epicentre, R1810M), and incubated at 37 °C for 1.5 h. SDS was added to the cell suspension to a final concentration of 1% to solubilize the chromatin. A volume of 150 µL of TES (50 mM Tris pH 8.0, 10 mM EDTA, 1% SDS) was added to the solution and the cell lysate was incubated at 65 ˚C to reverse the crosslinks. After reversal of crosslinks, the WGS samples were processed and sequenced in the same way as ChIP-seq samples above. WGS of B31-derived strain S9 (see Supplementary Data 1) was performed on cells grown and treated similarly to the ones used for the ChIP experiments described above.

Strain CJW_Bb523 was grown in 40 mL BSK-II medium to a density of ~3–6 × 10^7 cells/mL. The culture was pelleted at 4000 × $g$ for 10 min in the swinging bucket rotor, resuspended in 1 mL cold HN buffer, and re-pelleted. Genomic DNA was then extracted using DNeasy Blood and Tissue Kits (Qiagen) following the manufacturer's recommendation for Gram-negative bacteria. Library preparation and whole-genome sequencing were done by the Yale Center for Genome Analysis on a NovaSeq6000 instrument with 2 × 150 bp read length.

## Sequence mapping and analysis

The sequencing reads for ChIP-seq and WGS were mapped using CLC Genomics Workbench (CLC Bio, Qiagen) to the combined *B. burgdorferi* genome (NCBI GCA_000008685.2_ASM868v2), modified as described below and in Supplementary Data 2, Worksheet 2. For each strain, the ChIP-seq reads were mapped to a genome sequence carrying the appropriate genetic modifications. Strains CJW_Bb378, CJW_Bb379, CJW_Bb403, CJW_Bb488, CJW_Bb519, CJW_Bb520, CJW_Bb524, CJW_BB525, and CJW_Bb610 have a P*flaB*-*aphI*-*flaB*t kanamycin resistance cassette replacing the intergenic region between *parZ* and *parB*. Additionally, they carry a translational fusion to one of the Par proteins, as well as, in some cases, deletion of a *par* gene. Since the binding of ParB and ParZ fusions to the *par* locus of these genetically modified strains also involved binding to the P*flaB* and *flaB*t sequences inserted into the *par* locus during strain generation, reads associated with these sequences were erroneously mapped to the

endogenous P$_{flaB}$ and *flaB*t sequences located in the *flaB* locus. To circumvent this problem, we removed the P$_{flaB}$ and *flaB*t sequences from the *flaB* locus of the genome sequences prior to ChIP-seq read mapping. Since these ChIP-seq strains also contain a Δ*bbe02*::P$_{flaB}$-*aadA* genetic modification on lp25, we also removed the P$_{flaB}$ sequence from this modified lp25 sequence prior to read mapping. For the same reasons, prior to read mapping, we removed the chromosomal sequences P$_{O826}$ and P$_{flgB}$, in strain CJW_Bb101, which carries P$_{O826}$ and P$_{flgB}$ close to the *parZ* insert on the shuttle vector. Similarly, to prevent erroneous read mapping, we removed the *parB* sequence from the shuttle vector in strain CJW_Bb403. We note that this shuttle vector lacks a *parS* site. In the absence of *parS*, mCherry-ParB is not expected to bind the shuttle vector *parB* sequence significantly, as evidenced by the absence of fluorescent puncta formation (Supplementary Fig. 1b). For all ChIP-seq samples, final mapping was done to a concatenated genome sequence obtained by linking the chromosomal sequence, modified as described above, to the sequences of the plasmids present in each strain. Additionally, since shuttle vectors are found in five-fold higher copy number than the chromosome[22,23,63], for strains containing shuttle vectors we included 5 tandem copies of the shuttle vector in the concatenated genome prior to read mapping, then summed the binding to these copies.

Sequencing reads from each ChIP and WGS samples were normalized by the total number of reads. The WGS results were used as the "input" control for ChIP-seq samples. For marker frequency analysis, the reads corresponding to the *oriC* and *terC* regions were averaged over a 30 kilobase span. For the plasmids, the reads were averaged over the entire size of each plasmid. The ChIP enrichment (ChIP/input) and the locus ratios were plotted and analyzed using R scripts, which are available from https://github.com/xindanwanglab/takacs-2022-natcomm[101]. For the whole genome sequencing of strain CJW_Bb523, the sequencing reads were processed using Trimmomatic[102] (parameter = PE, -phred33, -baseout, ILLUMINACLIP:2:30:10, TRAILING:20, MINLEN: 36) and around 8 million reads were mapped to *B. burgdorferi* B31 genome using Bowtie2[103] (parameter = -non-deterministic). HTSeq[104] was used to obtain the genomic coverage. The normalized count for each plasmid was calculated as the sum of reads at every base pair (bp) divided by the plasmid size. For *oriC*, we only considered reads mapped to nucleotide position 443,037-473,267 (~30 kb region, similar to the average size of *Borrelia* plasmids) on the main chromosome.

## DNA fluorescence in situ hybridization (FISH)

*B. burgdorferi* cells were washed and resuspended in 1x PBS prior to the FISH experiments in order to remove contaminants from the BSK-H/BSK-II media. Cells were placed on poly-L-lysine coated wells that were outlined on coverslips with a hydrophobic pen (Super PAP pen, Invitrogen #008899). Prior to use, the coverslips were cleaned by sonication steps in 1 M KOH, miliQ water, and 70% ethanol alternated with triple milliQ water rinses, as described previously[41]. Once applied to the poly-L-lysine coated wells, *B. burgdorferi* cells were fixed in 4% formaldehyde in PBS for 5 min at RT. Cells were then washed 3 times in PBS. Cells were permeabilized using 400 µg/mL lysozyme in GTE buffer (50 mM glucose, 25 mM Tris, 10 mM EDTA, pH 8.0), then washed 3 times with PBS. A hybridization buffer was adapted from a previous report[105]. Briefly, 1 g dextran sulfate was dissolved in 5 mL dH2O, then 3530 µL formamide, 10 mg *E. coli* tRNA, 1 mL 20× SSC buffer (175.3 g/L NaCl, 88.2 g/L sodium citrate, pH 7.0), and 80 µL of 25 mg/mL BSA were added. The hybridization buffer was filtered and stored at -20 °C. Cells, after fixation and permeabilization, were pre-hybridized in hybridization buffer containing 1 mg/mL RNase A for 30 min. Then, the cells were denatured on a heat block at 94 °C for 5 min. First, the cells were placed on the heat block for 1 min in pre-hybridization solution, which was then replaced with hybridization solution, containing 200 nM probe. The Stellaris locked nucleic acid oligonucleotide FISH probe (sequence 5'-

GAATAAGTAAAAGTGGTTTAG-3', labeled with the dye 6FAM) was synthesized at Biosearch Technologies. This probe recognized a highly repetitive sequence present in 176 copies on plasmid lp21 of strain B31 and in the right subtelomeric region of the chromosome of strain 297[18,106]. The large number of repeats ensured that a strong fluorescent signal could be achieved despite using a single fluorescent probe. Strain B31e2[107], which no longer harbors plasmid lp21 of the parental strain B31, served as a negative control as it lacks the target sequence of FISH probe. Hybridization proceeded at RT for 2 h. Wells were then washed as follows: 3 × 10 min washes with 40% (wt/vol) formamide in 2× SSC, 2 × 5 min washes in 2x SSC, and 3× washes in 1× PBS. Prior to imaging, 1 mg/mL DAPI in SlowFade® Gold Antifade Mountant (Thermo Fisher #S36937) was applied to each well.

## RNA isolation, qRT-PCR, and qPCR

For RNA isolation, exponentially growing *B. burgdorferi* cultures were diluted to $2 \times 10^5$ cells/mL in 30 mL BSK-II, grown in duplicate for 2 days, then RNA was extracted and quantified as previously described[85]. The cells were pelleted using a 10 min centrifugation at $4300 \times g$ in a swinging bucket rotor. The pellet was resuspended in 400 µL buffer HN[98] (50 mM NaCl, 10 mM HEPES, pH 8.0). RNA was stabilized using the RNAprotect bacteria reagent (Qiagen), extracted using enzymatic lysis and proteinase K digestion (protocol 4 in the RNAprotect bacteria reagent manual), and purified using the RNeasy minikit (Qiagen). DNA was removed using the Turbo DNA-free kit (Thermo Fisher Scientific). RNA was quantified using the Kapa SYBR Fast one-step qRT-PCR mastermix kit (Roche) using 10 ng total RNA per reaction, in duplicate. One additional reaction was performed for each sample without the reverse transcriptase and confirmed that the measured amplification was not due to DNA contamination of the RNA samples. The primers used for amplification of the *parZ* transcript were 5'-CCCCCTATTTTAAAAACCGAAG-3' and 5'-TAATGGTTTGCG CGTATCC-3'. The primers used for amplification of the control *recA* transcript were previously described[19]. The PCR cycling conditions used on a Bio-Rad CFX Connect Real-time system were: reverse transcription (5 min at 42 °C); enzyme activation (5 min at 95 °C); 40 cycles of annealing, extension, and data acquisition (3 sec at 95 °C and 20 s at 60 °C with fluorescence acquisition in the SYBR scan mode); and a melt curve analysis (55 to 95 °C in 0.05 °C increments). The amount of the *parZ* transcript was normalized to the level of the *recA* transcript and then expressed for each sample relative to mean levels measured in strain CJW_Bb488 using the $\Delta\Delta C_T$ method, as previously described[108].

For the qPCR analysis of genome copy numbers per cell, an exponentially growing culture of strain CJW_Bb339 was diluted to $10^4$ cells/mL in triplicate, then culture densities were measured by darkfield counting and samples were harvested at 3, 5, and 7 days. Briefly, 1 mL of culture was centrifuged for 10 min at $10,000 \times g$ and RT. The medium was aspirated and the pellet was resuspended in 1 mL cold buffer HN. The cells were pelleted again for 10 min at $10,000 \times g$ and the pellet was resuspended in 100 µL water, then heat inactivated for 5 min at 99 °C. The volume was brought up to 1 mL by addition of 900 µL of water and the samples were stored at −80 °C. For generation of the standard curves, gBlock DNA fragments identical in sequence to nucleotides 126,973 to 127,294 and 148,101 to 148,426 of the chromosome of *B. burgdorferi* strain B31 were synthesized at Integrated DNA Technologies. These fragments encompass the sequences amplified by the *recA* and *flaB*-specific qPCR primers[19,109], respectively, plus 50 additional bases on each side. The synthesized fragments were resuspended in water to a stock concentration of $10^{10}$ molecules/µL, then 10-fold serial dilutions in water were performed to generate standards of known concentrations. qPCR reactions were set-up using the Kapa SYBR Fast one-step qRT-PCR mastermix kit (Roche) without the reverse transcriptase. A total reaction of 20 µL was setup, which included 1 µL of standard (run in duplicate) or of sample (run in quadruplicate). The PCR conditions were identical to those for the

qRT-PCR assay described above except that the reverse transcription step was omitted. DNA concentrations of the samples were calculated based on the standard curve using the CFX Maestro Software (Bio-Rad).

## Phylogenetic analyses

Protein sequences were retrieved from the NCBI databases by Standard Protein BLAST searches using *B. burgdorferi* proteins as queries in the NCBI BLASTP web-based platform. For Borreliaceae ParZ sequence alignments and phylogenetic tree building and for *Brachyspira*, *Leptospira*, and *Treponema* ParB sequence alignments, only non-redundant RefSeq protein sequences were analyzed. For *Leptospira*, the analysis of ParB sequences was limited to those encoded by chromosome I. For ParZ-like phylogenetic tree building, all BLAST hits retrieved using *B. burgdorferi* ParZ as query were utilized. The BLAST search was performed across the three domains of life and among virus sequences. Sequences were aligned and the phylogenetic trees were built using the Geneious R10.0.9 sequence analysis software.

## Statistics and reproducibility

Statistical analyses were performed in GraphPad Prism 9.3.1 software or in Microsoft Excel 365. With the exception of the tick imaging experiment (Fig. 3), where one tick was processed for imaging, all other imaging experiments were performed at least twice. The images shown in Figs. 1a, 1c, 1d, 1f, 2a, 7a, 7b, 7d, 7e, 7f, 7h, 8a, 8b, 8c, 9a, 9d and in Supplementary Fig. 1b, 1c, 1d, 1g, 4f, 4g, 6c, 6f are therefore representative of at least two independent imaging instances of the same strain. The images shown in Fig. 4a for strain CJW_Bb474 are representative for that strain, and for strain CJW_Bb379, which was imaged under the same experimental conditions. Strains CJW_Bb474 is derived from strain CJW_Bb379 and expresses free GFP which does not affect the *oriC* copy number or density (Fig. 1b). Statistical summaries for all figure panels that involve data quantification are provided in Supplementary Data 2, Worksheet 3, and include the following information: strain used in the figure, number of imaging samples analyzed, and total number of cells for the combined replicates.

## Data visualization

Data visualization was achieved using the following software: MATLAB R2019a, GraphPad Prism 9.3.1, FiJi[96], and Adobe Illustrator 2023.

## Plasmid construction methods

Plasmids are listed in Supplementary Data 1, Worksheet 2. Oligonucleotide primers used in the plasmid generation process are listed in Supplementary Data 1, Worksheet 3. Requests for the plasmids generated in this study should be directed to Christine Jacobs-Wagner.

## I. Shuttle vectors

**pBSV2G_P$_{0826}$-mCherry$^{Bb}$-ParB.** The following fragments were assembled (using intermediary constructs) between the SacI and PstI sites of plasmid pBSV2G_2: (a) the promoter of *B. burgdorferi* gene *bb0826* (P$_{0826}$)[89], corresponding to bp 535523-535703 of strain B31's chromosome, PCR-amplified with primers NT115 and NT116 and digested with SacI and BamHI; (b) the *mcherry$^{Bb}$* sequence[89], amplified with primers NT100 and NT101 and digested with KpnI and BamHI; (c) *parB* (*bb0434*), amplified from strain B31's genomic DNA with primers NT230 and NT232 and digested with SalI and PstI.

**pBSV2G_P$_{0826}$-RBS-ParZ-msfGFP$^{Bb}$.** The following fragments were assembled (using intermediary constructs) between the SacI and KpnI and between the PstI and HindIII sites of plasmid pBSV2G_2, respectively: (a) promoter P$_{0826}$[89], flanked by SacI and BamHI restriction enzyme sites; (b) *parZ* (*bb0432*), PCR-amplified from strain B31's genomic DNA with primers NT363 and NT364 and digested with

BamHI and KpnI; (c) the *msfgfp$^{Bb}$* sequence[89], PCR-amplified with primers NT159 and NT160 and digested with PstI and HindIII.

**pKFSS1_P$_{0826}$-mCherry$^{Bb}$-ParB.** P$_{0826}$-*mcherry$^{Bb}$-parB* was moved from pBSV2G_P$_{0826}$-mCherry$^{Bb}$-ParB into pKFSS1_2 using SacI and HindIII.

**pKFSS1_P$_{0826}$-msfGFP$^{Bb}$-ParB.** *msfgfp$^{Bb}$* was PCR amplified using primers NT341 and NT162, digested with BamHI and KpnI and ligated into the BamHI/KpnI backbone of pKFSS1_P$_{0826}$-mCherry$^{Bb}$-ParB.

**pBSV2G_P$_{0031}$-mCherry$^{Bb}$-ParB$^{P1}$.** The following fragments were assembled, through intermediates, between the SacI/KpnI, and XbaI/HindIII sites of pBSV2G_2: promoter P$_{0031}$[89], obtained from strain B31's genomic DNA by amplification with primers NT111 and NT112 and digestion with SacI and BamHI; *mcherry$^{Bb}$*, PCR amplified with NT100 and NT101 and digested with BamHI and KpnI; and *parB$^{P1}$*, obtained by de novo gene synthesis at Genewiz flanked by XbaI and HindIII restriction enzyme sites. The *parB$^{P1}$* gene encodes the P1 plasmid ParB protein minus its N-terminal peptide sequence (ParB$^{\Delta N30}$)[110], which was codon-optimized for translation in *B. burgdorferi* using the web-based JAVA Codon Adaptation tool hosted at www.jcat.de[111] as previously described[89] and was deposited at GenBank under accession number ON321895.

**pBSV2G_P$_{0031}$-msfGFP$^{Bb}$-ParB$^{P1}$.** *msfgfp$^{Bb}$* was PCR amplified using primers NT161 and NT162, digested with BamHI and KpnI, and cloned into the BamHI/KpnI sites of pBSV2G_P$_{0031}$-mCherry$^{Bb}$-ParB$^{P1}$.

**pBSV2B_P$_{0826}$-mCherry$^{Bb}$-ParB.** P$_{0826}$-*mcherry$^{Bb}$-parB* was moved from pBSV2G_P$_{0826}$-mCherry$^{Bb}$-ParB into pBSV2B using SacI and HindIII digestion.

**pBSV2B_r(P$_{0031}$-msfGFP$^{Bb}$-ParB$^{P1}$)_P$_{0826}$-mCherry$^{Bb}$-ParB.** P$_{0031}$-*msfgfp$^{Bb}$-parB$^{P1}$* was released from pBSV2G_P$_{0031}$-msfGFP$^{Bb}$-ParB$^{P1}$ as a SacI/FspI fragment and was ligated into the SacI/BsrBI backbone of pBSV2B_P$_{0826}$-mCherry$^{Bb}$-ParB.

**pBSV2G_P$_{smcL}$-mCherry$^{Bb}$-Smc.** The following fragments were sequentially assembled within the multicloning site of plasmid pBSV2G_2: *mcherry$^{Bb}$*, PCR-amplified using primers NT100 and NT101 and digested with BamHI and KpnI; a DNA fragment of 804 bp upstream of the ttg START codon of *smc* (*bb0045*), PCR-amplified with primers NT217 and NT219 and digested with SacI and BamHI; and a DNA fragment encoding *smc* (*bb0045*), PCR-amplified using primers NT221 and NT222 and digested with SalI and PstI.

**pBSV2G_P$_{0826}$-ParA-msfGFP$^{Bb}$.** The following fragments were assembled between the SacI and HindIII sites of pBSV2G_2: (i) promoter P$_{0826}$[89], flanked by SacI and BamHI restriction enzyme sites; (ii) msfGFP$^{Bb}$, obtained by PCR amplification using primers NT159 and NT160 and flanked by PstI and HindIII restriction enzyme sites; and *parA* (*bb0431*), obtained by PCR amplification using primers NT345 and NT226 and digestion with BamHI and KpnI.

**pBSV2H_P$_{0826}$-ParA-msfGFP$^{Bb}$.** P$_{0826}$-ParA-msfGFP$^{Bb}$ was transferred from pBSV2G_P$_{0826}$-ParA-msfGFP$^{Bb}$ into pBSV2H using BsrBI and AvrII.

## II. Suicide vectors

Note on nomenclature: "pKI" signifies a suicide vector generated to mediate knock-in of a gene of interest into the *B. burgdorferi* genome. While most of these plasmids are based on pCR2.1, some are based on other backbones. The antibiotic resistance used for selection is noted in the name of the construct (i.e. pKIKan, pKIGent, or pKIStrep) and is driven by default by the P$_{flgB}$ promoter. When, instead, the P$_{flaB}$

promoter was used to drive antibiotic resistance gene expression, this is noted in the name of the plasmid (e.g., pKIKan(P$_{flaB}$)).

**pKIGent.** This plasmid, obtained through a series of intermediate constructs, contains the following features: a) a $\Delta aphI\Delta bla$ backbone of plasmid pCR2.1[85], flanked by HindIII and XbaI restriction enzyme sites; b) a P$_{flgB}$-*aacC1*-*flaB*t cassette flanked by SpeI and SacII restriction enzyme sites. This cassette is flanked by two multi-cloning sites, namely HindIII-KpnI-SacI-BamHI-SpeI-XmaI and SacII-EcoRI-PstI-NotI-XhoI-SphI-XbaI. To generate this cassette, the following fragments were assembled through intermediates: (i) P$_{flgB}$-*aacC1*[87] was cloned in between the XmaI and the SacII sites of the backbone; (iii) *flaB*t, the flagellin transcriptional terminator, was generated by annealing primers NT350 and NT351 and ligating them into the SpeI/XmaI sites of the backbone. This inactivated the original XmaI site of the backbone but generated a new one downstream of *flaB*t.

**pKIGent_parS$^{P1}$.** This plasmid differs from pKIGent, in that the *parS$^{P1}$* sequence[112] was synthesized de novo at Genewiz and cloned into the pUC57Amp backbone. It was then PCR-amplified using primers NT165 and NT166, digested with SpeI and XmaI, and inserted into the SpeI/XmaI sites of the backbone. The resulting clone had a mutation in the *parS$^{P1}$* sequence that was corrected by site-directed mutagenesis using primers NT215 and NT216.

**pKIGent_parS$^{P1}$_phoU.** *B. burgdorferi* B31 chromosomal region from nucleotide 38650 to 40720 was PCR-amplified with primers NT175 and NT176 and digested with SacI and SpeI. The chromosomal region from nucleotide 40721 to 42797 was PCR-amplified with primers NT177 and NT178 and digested with PstI and XhoI. The two fragments were inserted into the SacI/SpeI and PstI/XhoI sites of pKIGent_parS$^{P1}$, respectively.

**pKIGent_parS$^{P1}$_cp26.** *B. burgdorferi* B31 cp26 region from nucleotide 20585 to 22602 was PCR-amplified with primers NT203 and NT204 and digested with SacI and BamHI. The cp26 region from nucleotide 22603 to 24632 was PCR-amplified with primers NT205 and NT206 and digested with PstI and XhoI. The two fragments were inserted into the SacI/BamHI and PstI/XhoI sites of pKIGent_parS$^{P1}$, respectively.

**pKIGent_parS$^{P1}$_uvrC.** *B. burgdorferi* strain B31's chromosomal region from nucleotide 474180 to 476218 was PCR-amplified with primers NT267 and NT268 and digested with SacI and SpeI. The region from nucleotide 476251 to 478279 was PCR-amplified with primers NT269 and NT270 and digested with PstI and XhoI. The two fragments were inserted into the SacI/SpeI and PstI/XhoI sites of pKIGent_parS$^{P1}$, respectively.

**pΔparBS.** *B. burgdorferi* chromosomal region between nucleotides 448842 and 450913 was PCR-amplified with primers NJ99 and NJ100 and digested with BamHI and XmaI. The region from nucleotide 452017 to 453998 was PCR-amplified using primers NJ97 and NJ98 and digested with PstI and XhoI. The two fragments were inserted into the BamHI/XmaI and PstI/XhoI sites of pKIGent, respectively.

**pΔparBS(Kan).** The P$_{flaB}$-*aphI* cassette from pKIKan(P$_{flaB}$) was excised using PstI and XmaI and inserted into the PstI/XmaI of pΔparBS.

**pKIGent_par.** The *B. burgdorferi* chromosomal region from nucleotide 448842 to 450913 was PCR-amplified with primers NJ99 and NJ100 and digested with BamHI and XmaI. The region between nucleotides 451133 and 453037 was PCR-amplified with primers NJ101 and NJ102 and digested with PstI and XhoI. The two fragments were inserted into the BamHI/XmaI and PstI/XhoI sites of pKIGent, respectively.

**pKIGent_parS$^{P1}$_lp17.** The 1.3 kilobase pair (kbp) BamHI/FspI fragment of pKIGent_parS$^{P1}$ was ligated with the 2.9 kbp BamHI/NaeI backbone of pKK81.

**pΔparAZ.** Nucleotides 447274 through 449320 of strain B31 chromosome were PCR-amplified using primers NT530 and NT531, digested with BamHI and XmaI, and ligated into the BamHI/XmaI backbone of pKIGent_par.

**pΔparZ.** Nucleotides 448134 through 450172 of the B31 chromosome were PCR-amplified using primers NT528 and NT529, cut with BamHI and XmaI, and ligated into the BamHI/XmaI backbone of pKIGent_par.

**pΔparAZBS.** Nucleotides 447274 through 449320 of the B31 chromosome were PCR-amplified using primers NT530 and NT531, digested with BamHI and XmaI, and ligated into the BamHI/XmaI backbone of pΔparBS.

**pKIGent_parS$^{P1}$_lp28-3.** Nucleotides 1853 to 3891 of plasmid lp28-3 were PCR-amplified using primers NT487 and NT488 and digested with KpnI and BamHI. Nucleotides 3890 to 5944 of plasmid lp28-3 were PCR-amplified using primers NT489 and NT490 and digested with SacII and XhoI. These fragments were cloned into the corresponding sites of pKIGent_parS$^{P1}$.

**pKIGent_parS$^{P1}$_lp28-2.** Plasmid lp28-2 was PCR-amplified using primers NT483 and NT484 and digested with KpnI and SpeI. Plasmid lp28-2 was PCR-amplified using primers NT485 and NT486 and digested with SacII and XhoI. These fragments were cloned into the corresponding sites of pKIGent_parS$^{P1}$.

**pKIGent_parS$^{P1}$_lp25.** The P$_{flgB}$-*aacC1*-*parS$^{P1}$* cassette was PCR-amplified from pKIGent_parS$^{P1}$ using primers NT509 and NT524, cut with PacI and MluI, and cloned into the PacI/MluI backbone of pKbeKan.

**pKIGent_parS$^{P1}$_lp38.** Nucleotides 20973 through 23014 of plasmid lp38 were PCR-amplified using primers NT499 and NT500, then digested with KpnI and SpeI. Nucleotides 23014 through 25091 of plasmid lp38 were PCR-amplified using primers NT501 and NT502, then digested with SacII and XhoI. The fragments were ligated into the corresponding sites of pKIGent_parS$^{P1}$.

**pKIGent_parS$^{P1}$_lp36.** Plasmid lp36 was PCR-amplified with primers NT495 and NT496, digested with KpnI and BamHI, then cloned into the KpnI/BamHI sites of pKIGent_parS$^{P1}$. The resulting plasmid was PCR-amplified using primers MRS_17 and MRS_18 and Gibson-assembled with a PCR product obtained by amplification of plasmid lp36 using primers MRS_19 and MRS_20.

**pKIKan.** This plasmid was obtained through a series of intermediate constructs, and in a manner similar to pKIGent, except that is has an *aphI* kanamycin resistance gene instead of an *aacC1* gentamicin resistance gene under the control of P$_{flgB}$. It contains the following features: a) the $\Delta aphI\Delta bla$ backbone of plasmid pCR2.1[85], flanked by HindIII and XbaI restriction enzyme sites; b) a P$_{flgB}$-*aphI*-*flaB*t cassette flanked by SpeI and SacII restriction enzyme sites that are part of two multicloning sites: HindIII-KpnI-SacI-BamHI-SpeI-XmaI and SacII-PstI-NotI-XhoI-SphI-XbaI. To generate this cassette, the following fragments were assembled through intermediates: (i) P$_{flgB}$, flanked by SacII and NdeI sites; (ii) *aphI*, PCR-amplified from pBSV2 using primers NT534 and NT535, digested with NdeI and XmaI, and cloned together with P$_{flgB}$ between the SacII and XmaI sites of the backbone; and (iii) *flaB*t, generated by annealing primers NT350 and NT351 and ligating them into the SpeI/XmaI sites of the backbone. This inactivated the

original XmaI site of the backbone but generated a new one downstream of *flaB*t.

**pKIKan(P$_{flaB}$)**. P$_{flaB}$ was PCR-amplified from pBSV2G_P$_{flaB}$-mCerulean$^{Bb}$ using primers NT577 and NT578, digested with SacII and NdeI, and ligated into the SacII/NdeI backbone of pKIKan.

**pKIKan_parS$^{P1}$**. P$_{flgB}$-*aphI-flaB*t was moved from pKIKan into the backbone of pKIGent_parS$^{P1}$ as an XmaI/PstI fragment.

**pKIKan(P$_{flaB}$)_parS$^{P1}$**. The P$_{flaB}$-*aphI-flaB*t cassette was moved from pKIKan(P$_{flaB}$) into the backbone of pKIGent_parS$^{P1}$ as an XmaI/PstI fragment.

**pKIKan_parS$^{P1}$_uvrC**. P$_{flgB}$-*aphI-parS$^{P1}$* was excised from pKIKan_parS$^{P1}$ using SpeI and PstI and inserted into the SpeI/PstI backbone of pKIGent_parS$^{P1}$_uvrC, where it replaced P$_{flgB}$-*aacC1-parS$^{P1}$*.

**pKIGent_parS$^{P1}$_lp28-4**. The pKIGent_parS$^{P1}$ backbone was PCR-amplified using primers MRS_46 and MRS_31. Plasmid lp28-4 was PCR-amplified using primers MRS_49 and MRS_53, and MRS_51 and MRS_52, respectively. The P$_{flgB}$-*aacC1-parS$^{P1}$* cassette was PCR-amplified from pKIGent_parS$^{P1}$ using primers MRS_32 and MRS_33. The four PCR products were Gibson assembled together.

**pKIGent_parS$^{P1}$_lp54**. pABA01 was PCR-amplified using primers MRS_24 and MRS_29. The P$_{flgB}$-*aacC1-parS$^{P1}$* cassette was PCR-amplified from pKIGent_parS$^{P1}$ using primers MRS_25 and MRS_26. The two PCR products were Gibson assembled.

**pΔparS**. A region between nucleotides 449961 and 451998 of strain B31's chromosome was PCR-amplified using primers NT625 and NT626, digested with BamHI and XmaI, and inserted into the BamHI/XmaI sites of plasmid pΔparBS.

**pΔparB**. A region from nucleotide 451996 to nucleotide 453998 of strain B31's chromosome was PCR amplified using primers NT627 and NT628, digested with PstI and XhoI, and inserted into the PstI/XhoI sites of pΔparBS.

**pΔparA(Kan)**. *parA* was deleted from pKIKan(P$_{flaB}$)_par by site-directed mutagenesis using primers NT623 and NT624.

**pKIKan(P$_{flaB}$)_par**. The P$_{flaB}$-*aphI-flaB*t cassette from pKIKan(P$_{flaB}$) was excised using PstI and XmaI and inserted into the PstI/XmaI sites of pKIGent_par.

**pKIKan(P$_{flaB}$)_ParZ-msfGFP$^{Bb}$**. *msfgfp$^{Bb}$* was PCR-amplified with primers NT643 and NT644. Plasmid pKIKan(P$_{flaB}$)_par was PCR-amplified with primers NT645 and NT646. The two fragments were Gibson-assembled.

**pKIKan(P$_{flaB}$)_mCherry$^{Bb}$-ParB**. *mcherry$^{Bb}$* was PCR-amplified with primers NT629 and NT630. Plasmid pKIKan(P$_{flaB}$)_par was PCR-amplified with primers NT631 and NT632. The two fragments were Gibson-assembled.

**pKIGent_parS$^{P1}$_lp28-1**. P$_{flgB}$-*aacC1-parS$^{P1}$* was PCR-amplified from pKIGent_parS$^{P1}$ using primers NT762 and NT763. The backbone of p28-1::flgBp-aacC1 was PCR-amplified using primers NT764 and NT765. The two fragments were Gibson-assembled together.

**pKIKan(P$_{flaB}$)_ParA-msfGFP$^{Bb}$**. *msfgfp$^{Bb}$* was PCR-amplified using primers NT633 and NT634. *parA* was PCR-amplified using primers NT636 and NT768. *parZ* was PCR-amplified from pKIKan(P$_{flaB}$)_par using primers NT766 and NT635. The suicide vector backbone was PCR-amplified from pKIKan(P$_{flaB}$)_par using primers NT769 and NT767. The four PCR products were Gibson-assembled together.

**pKIKan(P$_{flaB}$)_ParZ-msfGFP$^{Bb}$_ΔparA**. Site-directed mutagenesis was performed on plasmid pKIKan(P$_{flaB}$)_ParZ-msfGFP$^{Bb}$ using primers NT623 and NT624.

**pKIKan(P$_{flaB}$)_ParA-msfGFP$^{Bb}$_ΔparZ**. Site-directed mutagenesis was performed on plasmid pKIKan(P$_{flaB}$)_ParA-msfGFP$^{Bb}$ using primers NT778 and NT779.

**pKIKan(P$_{flaB}$)_ΔparBS_ParA-msfGFP$^{Bb}$**. The BamHI/XmaI insert of plasmid pKIKan(P$_{flaB}$)_ParA-msfGFP$^{Bb}$ was ligated into the BamHI/XmaI backbone of plasmid pΔparBS(Kan).

**pKIKan(P$_{flaB}$)_ΔparBS_ParZ-msfGFP$^{Bb}$**. The BamHI/XmaI insert of plasmid pKIKan(P$_{flaB}$)_ParZ-msfGFP$^{Bb}$ was ligated into the BamHI/XmaI backbone of plasmid pΔparBS(Kan).

**pKIKan(P$_{flaB}$)_mCherry$^{Bb}$-ParB_ΔparAZ**. The BamHI/XmaI insert of plasmid pΔparAZ was ligated into the BamHI/XmaI backbone of plasmid pKIKan(P$_{flaB}$)_mCherry$^{Bb}$-ParB.

**pKIKan(P$_{flaB}$)_ParA-msfGFP$^{Bb}$_ParZ$^{ΔN20}$**. Site-directed mutagenesis was performed on pKIKan(P$_{flaB}$)_ParA-msfGFP$^{Bb}$ using primers NT1020 and NT1021.

**pKIGent_parS$^{P1}$_lp21_V2**. The backbone of pKIGent_parS$^{P1}$ was PCR-amplified with primers NT800 and NT801. The P$_{flgB}$-*aacC1_parS$^{P1}$* cassette was PCR-amplified from pKIGent_parS$^{P1}$ with primers NT796 and NT797. Nucleotides 253 through 1114 of the B31 plasmid lp21 were PCR-amplified using primers NT794 and NT795. Nucleotides 1115 through 2628 of the B31 plasmid lp21 were PCR-amplified using primers NT798 and NT799. The four PCR products were Gibson-assembled.

**pKIGent_parS$^{P1}$_lptD**. The backbone of pKIGent_parS$^{P1}$ was PCR-amplified with primers NT824 and NT825. The P$_{flgB}$-*aacC1_parS$^{P1}$* cassette was PCR-amplified from pKIGent_parS$^{P1}$ with primers NT820 and NT821. Nucleotides 895077 through 896572 of the B31 chromosome were PCR-amplified using primers NT818 and NT819. Nucleotides 896573 through 898070 of the B31 chromosome were PCR-amplified using primers NT822 and NT823. The four PCR products were Gibson-assembled.

**pKIGent_mCherry$^{Bb}$-Smc**. The following fragments were Gibson-assembled: 1) nucleotides 43811 through 45353 of the B31 chromosome fused downstream of and in frame to *mcherry$^{Bb}$*, controlled by the *smc* native promoter, obtained by PCR-amplification of plasmid pBSV2G_P$_{smcL}$-mCherry$^{Bb}$-Smc with primers NT1006 and NT1007; 2) the gentamicin cassette of pKIGent_parS$^{P1}$_phoU, obtained by PCR-amplification with primers NT1008 and NT1009; 3) nucleotides 45376 through 46530 of the B31 chromosome, obtained by PCR-amplification with primers NT1010 and NT1011; and 4) the suicide vector backbone of plasmid pΔparA(kan), obtained by PCR-amplification with primers NT1012 and NT1013.

**pΔsmc(gent)**. The following fragments were Gibson-assembled: 1) nucleotides 42016 through 43530 of the B31 chromosome, obtained by PCR amplification with primers NT960 and NT961; 2) the gentamicin cassette of pKIGent_parS$^{P1}$_phoU, obtained by PCR amplification with primers NT962 and NT963; 3) nucleotides 45332 through 46857 of the B31 chromosome, obtained by PCR amplification with primers NT964 and NT965; and 4) the suicide vector backbone of plasmid

pΔparA(kan), obtained by PCR amplification with primers NT966 and NT967.

**pKIGent_ΔparBS_ParZ-msfGFP^Bb_ΔparA.** The gentamicin resistance cassette was excised from pΔparB as a SacII/XmaI fragment and inserted into the SacII/XmaI backbone of plasmid pKIKan(P_flaB)_ΔparBS_ParZ-msfGFP^Bb_ΔparA.

**pKIKAN(P_flaB)_ΔparBS_ParZ-msfGFP^Bb_ΔparA.** The BamHI/XmaI insert of plasmid pKIKan(P_flaB)_ParZ-msfGFP^Bb_ΔparA was ligated into the BamHI/XmaI backbone of plasmid pΔparBS(Kan).

**pKIKAN(P_flaB)_mCherry^Bb-ParB_ΔparZ.** The BamHI/XmaI insert of plasmid pΔparZ(Kan) was ligated into the BamHI/XmaI backbone of plasmid pKIKan(P_flaB)_mCherry^Bb-ParB.

**pΔparZ(Kan).** The P_flaB-aphI cassette of pKIKan(P_flaB) was excised using PstI and XmaI and ligated to the PstI/XmaI backbone of plasmid pΔparZ.

**pKIStrep(P_flaB)_parS^P1.** aadA was PCR-amplified from pJSB252 using with primers NT756 and NT757. The backbone of the pKIKan(P_flaB)_parS^P1 plasmid was PCR-amplified using primers NT758 and NT759. The two fragments were Gibson-assembled with each other.

**pKIStrep_parS^P1_Tn.** pGKT was digested with EagI, and the resulting 1.8 kbp fragment was purified and ligated, yielding plasmid pTG, which was then PCR-amplified using primers NT690 and NT691. P_flaB-aadA-parS^P1 was PCR-amplified from pKIStrep(P_flaB)_parS^P1 using primers NT692 and NT693. The two PCR products were Gibson assembled with each other. The resulting plasmid was PCR amplified with NT784 and NT785 and the resulting PCR product was digested with AvrII and self-ligated.

### Reporting summary
Further information on research design is available in the Nature Portfolio Reporting Summary linked to this article.

## Data availability
The reference *B. burgdorferi* B31 genome is available from NCBI (GenBank assembly accession code GCA_000008685.2). The sequence of the codon-optimized *parB^P1* gene was deposited with Genbank (accession code ON321895). The ChIP-Seq and WGS data generated in this study are deposited in the NCBI Gene Expression Omnibus platform and are publicly available through GEO Series accession number GSE202255. The accession numbers for each sample can be found in Supplementary Data 2, Worksheet 2. Requests for other raw data should be directed by email to the corresponding authors. Source data are provided with this paper.

## Code availability
The image analysis code developed as part of this study was deposited in Github (https://github.com/JacobsWagnerLab/published)[113]. The R scripts used to perform and plot the ChIP enrichment (ChIP/input) and the marker frequency analyses were also deposited in Github (https://github.com/xindanwanglab/takacs-2022-natcomm)[101].

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

## Acknowledgements

The authors thank Stanford University's Cell Sciences Imaging Core Facility (CSIF, RRID:SCR_017787) for the use of its OMX BLAZE microscope, Stanford Medicine's Animal Histology Services for generating the thin cryosections of OCT-embedded tick midgut samples, Stanford's Center for Innovation in In vivo Imaging (SCi³) for use of its cryostat, and Indiana University Center for Genomics and Bioinformatics for deep sequencing. We are grateful to Drs. Brian Stevenson, Linda Bockenstedt, Erol Fikrig, Nikhat Parveen, Melissa Caimano, and Jon Blevins for sharing strains and plasmids, and David Rudner for anti-GFP and anti-mCherry antibodies. This project was supported in part by award number 1S10OD01227601 from the National Center for Research Resources (NCRR) to Stanford University's CSIF. C.N.T. was supported in part by an American Heart Association postdoctoral fellowship (award number 18POST33990330). X.W. was supported by National Institutes of Health grants R01GM141242 and R01GM143182. P.A.R. and J.W. were supported by the Intramural Research Program of the National Institute of Allergy and Infectious Diseases, National Institutes of Health. C.J.-W. is a Howard Hughes Medical Institute Investigator. The funders had no role in study design, data collection, analysis, and interpretation, or the decision to submit the work for publication.

## Author contributions

C.N.T. and C.J.-W. designed the research. J.W. and P.A.R. performed the tick-mouse transmission studies, tick dissections, and tick tissue preservation. C.N.T. generated plasmids and strains, imaged the samples, analyzed the images, performed phylogenetic analyses, and visualized the data. Y.X. developed the MATLAB image analysis pipeline with input from C.N.T. Z.R., X.K., and X.W. performed ChIP-seq and WGS analyses and visualization. I.I. performed WGS analyses. M.S. performed DNA FISH. M.R.S. and N.J. generated plasmids and strains. P.A.R. provided reagents. C.N.T., P.A.R., X.W., and C.J-W. supervised the project and acquired funding. C.N.T. and C.J-W. wrote the paper with input from all authors.

## Competing interests

The authors declare no competing interests.
