## [Peer Review File · Nature Communications]

Polyploidy, regular patterning of genome copies, and unusual control of DNA partitioning in the Lyme disease spirocheteReviewer #1 (Remarks to the Author):

The manuscript by Takacs et al sets out to answer one of the most enigmatic questions of *Borrelia* biology, which is how the pathogen faithfully segregates its highly segmented genome to progeny during cell division. To do this, the authors have adapted a fluorescence microscopy method involving expression of a fluorescent protein-tagged ParB protein that leads to a signal that pinpoints the chromosomal and plasmid loci within the spirochetal cell. The authors should be commended for the exhaustive measures taken to provide numerous, solid controls that clearly demonstrate the specificity and efficacy of their chosen experimental approach. Their findings demonstrate that the *B. burgdorferi* genome is polyploid during growth in culture, and show strong evidence that the chromosome and various plasmids exhibit a regular subcellular distribution within the bacterial cells. Moreover, the results show that the genome is polyploid during infection of the tick vector, demonstrating biological relevance. Finally, the authors present strong evidence that the subcellular distribution of the genome involves ParA that is controlled by a newly identified centromere-binding protein that they term ParZ. Although ParA is normally controlled by ParB, the study shows that ParB protein of *B. burgdorferi* is involved primarily in recruiting the condensing protein, SMC. Overall, this is a highly significant investigation of a poorly studied feature of *Borrelia* biology that will likely have an impact on future studies for the foreseeable future. This reviewer found few weaknesses, and the following comments are provided to further strengthen the manuscript.

Pg. 3, lines 61-64 – One noted exception to the referenced studies that showed *B. burgdorferi* plasmids are present in a roughly 1:1 ratio to the chromosome is a recently published study (Wong et al, 2022, *Front. Cell. Infect. Microbiol.*). The authors should note this difference and include the reference.

Pgs. 6-7, lines 130-134 – The authors state that the copy number of plasmids decreased as the in vitro culture advanced into stationary phase. However, the only data provided is for cp26. Were the copy numbers determined for the remaining 15 plasmids under study? This seems warranted given the differences in DNA topology between multiple *B. burgdorferi* plasmids (i.e., circular vs linear), and the prevalence for some to be routinely lost during culture. If the copy numbers have been determined, that information should be provided in the supplemental data section and briefly mentioned in the Results and Discussion sections.

Reviewer #2 (Remarks to the Author):

In this study, Takacs et al., investigate the pathways involved in genome inheritance in the spirochete *B. burgdorferi*. For this, they use a cell biological approach by labelling the endogenous ParB (as a proxy for the chromosomal oriC) as well as using orthogonal parS-ParB systems to label 16 plasmids present in this organism. In contrast to the prevailing model that suggests *B. burgdorferi* carries one chromosome per cell, Takacs et al., find that exponentially growing bacteria are polyploid and the number of oriC copies scales with the length of the cell linearly. They also establish that these cells have multiple ter copies and multiple plasmid copies as well. Interestingly, ori:ter ratio is ~ 1 , which suggests that replication likely occurs asynchronously. They find that oriC spacing is equidistant across the length of the cell. Takacs et al., further dissect the proteins required for ori segregation and copy number maintenance. They implicate ParA, ParB and a novel protein ParZ in orchestrating the same. They find that ParB binds parS sites near the ori and recruits SMC protein, which contributes mildly to ori segregation. ParB does not affect ParA localization. In contrast, they identify a second protein, ParZ that is essential for wild type ParA localization patterns. Together the ParA/ParZ and the ParB/SMC axis contribute to origin spacing; deletion of both arms results in an increase in anucleate cell production.

This is an excellent study, providing novel insights into an understudied and important pathogen, with universal implications on our understanding of bacterial genome inheritance mechanisms. The work is well-designed and well-executed. The writing is clear and conclusions are overall supported by the data. Indeed, the observations described here open several questions about the regulation of replication, stationary phase dynamics and the coordination of plasmid and chromosomal segregation. I appreciate that these are likely questions for future studies. Thus, I am in strong support of publication of this comprehensive body of work. However, I have a few queries that the authors should address prior to publication:

1. The authors reason that the discrepancy between the qPCR results and their present work on chromosome copy number could be due to the growth phase of the cells. Although they provide cell biological support for this (Fig. 1G), a qPCR measurement for copy number in exponential phase would provide a clearer resolution to the differences observed.
2. Authors should measure ParA expression levels in the parZ deletion or truncation background to rule out any impact of ParZ DNA binding on ParA expression (ChIP profiles suggest that ParZ binding extends into the parA gene as well).
3. What is the growth phenotype of the deletion mutants of parAZ, parB and parAB? Authors see a decrease in oriC copies as well as a mild increase in anucleate cells. However, do they observe growth defects in terms of doubling times, growth rates or CFUs for example?
4. Following from the question above, did the authors observe any impact of these chromosome segregation proteins on plasmid copy number? From the images in Fig. 2A, it appears that plasmids colocalize at some frequency with the oriC. Have the authors measured colocalization frequency? And does this and/ or copy number of plasmids get affected when origin spacing is perturbed?

Reviewer #3 (Remarks to the Author):

In this study, Takacs et al have employed a suite of tools, from fluorescence microscopy analysis, genetic modifications, to genome-wide ChIP-seq, to carefully investigate chromosome/plasmid distribution and partition in *B. burgdorferi*. Experiments revealed that *B. burgdorferi* is polyploidy during exponential growth and in ticks, and most excitingly, they discovered a novel centromere-binding protein called ParZ that together with ParABS-SMC ensures DNA partition and regular distribution of chromosomes in this bacterium. This is an important work for the field, I have read this work with much interest and just have a few comments for the authors to consider:

- I have not seen the numbers of anucleate cells for various strains being reported. Is this a sensible parameter to report for a polyploid organism? Naively, I thought the defect in chromosome spacing/partitioning must be manifested into a cost in the fitness of the organism i.e. a higher anucleate fraction and/or slow growth or delay/defect in cell division?
- The discovery of ParZ is very exciting and according to the ChIP-seq profile, ParZ seems to spread similarly to ParB. Since this is the most novel finding in this study, it would benefit the readers if Takacs et al could interpret the data and discuss it deeper. For examples, discussing the shape of the anti-ParZ ChIP-seq profile better. Discuss why there are several summits on the anti-ParZ ChIP-seq profile? Discuss why the ChIP-seq profiles of both ParB and ParZ drop significantly in the intergenic region between the two converging parZ and parB genes? And discuss the potential significance of ParZ spreading for the activation and interaction with ParA.
- Data in Supplementary figure 6E (ChIP-seq profile of ParZ with parZ on a shuttle plasmid) is not so convincing to show that the binding site of ParZ is within the parZ gene. There are several peaks and summits on that profile, and the peak within parZ is

not the most prominent one. A control of anti-ParZ ChIP-seq on an empty plasmid is perhaps required. It would be very convincing if Takacs et al could inspect the sequence immediately below the summit to identify any potential inverted repeat. No further experiment is required but some more analysis would strengthen the claim and add to the novelty of the ParZ protein.

- **Discuss the Myxococcus PadC-ParABS-SMC system if appropriate. This is another system that shows the plasticity in bacterial chromosome segregation.**
- **Table S2: indicate the antibodies used for ChIP-seq, anti-GFP/mCherry or anti-ParZ/ParB/SMC polyclonal antibodies etc.**

Response to the editor

Thank you for the opportunity to submit an amended version of our manuscript. We are delighted by the positive reviews and are very grateful to each reviewer for their time and helpful feedback. Please see below for a point-by-point response to each reviewer's comment (in blue).

REVIEWER COMMENTS

Reviewer #1 (Remarks to the Author):

The manuscript by Takacs et al sets out to answer one of the most enigmatic questions of *Borrelia* biology, which is how the pathogen faithfully segregates its highly segmented genome to progeny during cell division. To do this, the authors have adapted a fluorescence microscopy method involving expression of a fluorescent protein-tagged ParB protein that leads to a signal that pinpoints the chromosomal and plasmid loci within the spirochetal cell. The authors should be commended for the exhaustive measures taken to provide numerous, solid controls that clearly demonstrate the specificity and efficacy of their chosen experimental approach. Their findings demonstrate that the *B. burgdorferi* genome is polyploidy during growth in culture, and show strong evidence that the chromosome and various plasmids exhibit a regular subcellular distribution within the bacterial cells. Moreover, the results show that the genome is polyploidy during infection of the tick vector, demonstrating biological relevance. Finally, the authors present strong evidence that the subcellular distribution of the genome involves ParA that is controlled by a newly identified centromere-binding protein that they term ParZ. Although ParA is normally controlled by ParB, the study shows that ParB protein of *B. burgdorferi* is involved primarily in recruiting the condensing protein, SMC. Overall, this is a highly significant investigation of a poorly studied feature of *Borrelia* biology that will likely have an impact on future studies for the foreseeable future. This reviewer found few weaknesses, and the following comments are provided to further strengthen the manuscript.

Response: Thank you so much for the support. Much appreciated.

Pg. 3, lines 61-64 – One noted exception to the referenced studies that showed *B. burgdorferi* plasmids are present in a roughly 1:1 ratio to the chromosome is a recently published study (Wong et al, 2022, *Front. Cell. Infect. Microbiol.*). The authors should note this difference and include the reference.

Response: Thanks for bringing this paper to our attention. This study proposes a 3:1 ratio for lp17 whereas we found a 1.5:1 ratio. We have revised our manuscript to express a more nuanced comparison with the literature that includes this reference. Please see lines 135-137 in the revised main text.

Pgs. 6-7, lines 130-134 – The authors state that the copy number of plasmids decreased as the in vitro culture advanced into stationary phase. However, the only data provided is for cp26. Were the copy numbers determined for the remaining 15 plasmids under study? This seems warranted given the differences in DNA topology between multiple *B. burgdorferi* plasmids (i.e., circular vs linear), and the prevalence for some to be routinely lost during culture. If the copy numbers have been determined, that information should be provided in the supplemental data section and briefly mentioned in the Results and Discussion sections.

Response: In this study, we show the drop in copy number for the chromosome (and one important plasmid) only to provide an explanation for the discrepancy between our study (which demonstrates polyploidy) and the highly cited Morrison et al 1999 study (which suggests monoploidy at the chromosome level). We found that the data do actually agree when we control for the growth phase. Otherwise, the decline in genome copy in stationary phase is largely outside the scope of our study.

This said, we fully agree with the reviewer that a drop in plasmid copy number in stationary phase--which we indeed observed for all 16 tested plasmids--would be of great interest to the field for the reasons mentioned by the reviewer. In fact, we believe that this deserves its own follow-up study/paper in which we will describe the drop in plasmid copy number and its relevance in relation to bacterial physiology (including viability and other quantitative features) and the practicalities of growing *Borrelia* strains in axenic cultures. If this is acceptable to the reviewer, we think that presenting the plasmid data in this broader context will be more useful for the field.

Reviewer #2 (Remarks to the Author):

In this study, Takacs et al., investigate the pathways involved in genome inheritance in the spirochete *B. burgdorferi*. For this, they use a cell biological approach by labelling the endogenous ParB (as a proxy for the chromosomal oriC) as well as using orthogonal parS-ParB systems to label 16 plasmids present in this organism. In contrast to the prevailing model that suggests *B. burgdorferi* carries one chromosome per cell, Takacs et al., find that exponentially growing bacteria are polyploid and the number of oriC copies scales with the length of the cell linearly. They also establish that these cells have multiple ter copies and multiple plasmid copies as well. Interestingly, ori:ter ratio is ~ 1 , which suggests that replication likely occurs asynchronously. They find that oriC spacing is equidistant across the length of the cell. Takacs et al., further dissect the proteins required for ori segregation and copy number maintenance. They implicate ParA, ParB and a novel protein ParZ in orchestrating the same. They find that ParB binds parS sites near the ori and recruits SMC protein, which contributes mildly to ori segregation. ParB does not affect ParA localization. In contrast, they identify a second protein, ParZ that is essential for wild type ParA localization patterns. Together the ParA/ParZ and the ParB/SMC axis contribute to origin spacing; deletion of both arms results in an increase in anucleate cell production.

This is an excellent study, providing novel insights into an understudied and important pathogen, with universal implications on our understanding of bacterial genome inheritance mechanisms. The work is well-designed and well-executed. The writing is clear and conclusions are overall supported by the data. Indeed, the observations described here open several questions about the regulation of replication, stationary phase dynamics and the coordination of plasmid and chromosomal segregation. I appreciate that these are likely questions for future studies. Thus, I am in strong support of publication of this comprehensive body of work. However, I have a few queries that the authors should address prior to publication:

Response: We appreciate the kind comments and support. Thank you!

1. The authors reason that the discrepancy between the qPCR results and their present work on chromosome copy number could be due to the growth phase of the cells. Although they provide cell

biological support for this (Fig. 1G), a qPCR measurement for copy number in exponential phase would provide a clearer resolution to the differences observed.

Response: Good suggestion. A qPCR quantification is now provided as Supplementary Fig. 1h.

2. Authors should measure ParA expression levels in the *parZ* deletion or truncation background to rule out any impact of ParZ DNA binding on ParA expression (ChIP profiles suggest that ParZ binding extends into the *parA* gene as well).

Response: Good point. We now provide a quantitative comparison of ParA-msfGFP concentration and found only a small increase (~25-35%) in concentration in the *parZ* mutants relative to the parent strain. This small difference cannot account for the loss of the banded pattern observed in the *parZ* mutant strains, as the ParA-msfGFP banded localization pattern is not only retained but also require ParZ when ParA-msfGFP is expressed at 10-fold higher levels from a shuttle vector. This new information is now provided in Supplementary Figures 6a-c and lines 236-240.

3. What is the growth phenotype of the deletion mutants of *parAZ*, *parB* and *parAB*? Authors see a decrease in *oriC* copies as well as a mild increase in anucleate cells. However, do they observe growth defects in terms of doubling times, growth rates or CFUs for example?

Response: Yes, the $\Delta parAZBS$ mutant has slightly longer doubling times and a delay in colony formation. These data are now provided as Supplementary Fig. 6g and 6h and lines 346-349 in the revised manuscript.

4. Following from the question above, did the authors observe any impact of these chromosome segregation proteins on plasmid copy number? From the images in Fig. 2A, it appears that plasmids colocalize at some frequency with the *oriC*. Have the authors measured colocalization frequency? And does this and/or copy number of plasmids get affected when origin spacing is perturbed?

Response: Great questions! We have actually done a large Hi-C study of the genome in wild-type and various mutants that addresses some of these questions. We found that with the exception of a couple of plasmids, there is no evidence of interaction between plasmids and the chromosome. Even for the exceptions (for example, *lp21*), the interaction was weak, suggesting that it was quite transient. Interestingly, the interaction of these plasmids was with the origin region of the chromosome, and it decreased in the *parAZBS* mutant, suggesting a potential involvement of the chromosome segregation proteins. We feel that our Hi-C study, which includes data about the organization of this highly segmented genome and is currently in preparation for publication, is beyond the scope of this paper and would be best described in a separate, follow-up paper (rather than in the supplementary data section of this paper). We hope that this approach is acceptable to the reviewer.

Reviewer #3 (Remarks to the Author):

In this study, Takacs et al have employed a suite of tools, from fluorescence microscopy analysis, genetic modifications, to genome-wide ChIP-seq, to carefully investigate chromosome/plasmid distribution and

partition in *B. burgdorferi*. Experiments revealed that *B. burgdorferi* is polyploidy during exponential growth and in ticks, and most excitingly, they discovered a novel centromere-binding protein called ParZ that together with ParABS-SMC ensures DNA partition and regular distribution of chromosomes in this bacterium. This is an important work for the field, I have read this work with much interest and just have a few comments for the authors to consider:

Response: Thank you so much for the supportive comment.

- I have not seen the numbers of anucleate cells for various strains being reported. Is this a sensible parameter to report for a polyploid organism? Naively, I thought the defect in chromosome spacing/partitioning must be manifested into a cost in the fitness of the organism i.e. a higher anucleate fraction and/or slow growth or delay/defect in cell division?

Response: Good point. We have added data showing that the mutants have indeed a higher anucleate fraction (Supplementary Fig. 6e) and growth defects (Supplementary Fig. 6g and 6h).

- The discovery of ParZ is very exciting and according to the ChIP-seq profile, ParZ seems to spread similarly to ParB. Since this is the most novel finding in this study, it would benefit the readers if Takacs et al could interpret the data and discuss it deeper. For examples, discussing the shape of the anti-ParZ ChIP-seq profile better. Discuss why there are several summits on the anti-ParZ ChIP-seq profile? Discuss why the ChIP-seq profiles of both ParB and ParZ drop significantly in the intergenic region between the two converging *parZ* and *parB* genes? And discuss the potential significance of ParZ spreading for the activation and interaction with ParA.

Response: The presence of several summits in the ChIP-seq profile of ParB homologs has also been reported (Graham 2014 Genes Dev, or Tran, Le et al NAR 2018) although the molecular reasons remain unknown. We speculate that if ParZ forms a clamp and slides on the DNA like ParB, then transcription machinery might “push” ParZ along the DNA leading to accumulation of ParZ downstream the direction of transcription. We have added this discussion to the revised paper (lines 399-417).

As for “the potential significance of ParZ spreading for the activation and interaction with ParA”, we assume that the benefits are the same as for ParB spreading: having a high number of ParB/ParZ molecules increases the probability of interaction with ParA dimers. Furthermore, multi-site attachment ensures that the ParZ/ParB partition complex remains in contact to the DNA matrix via at least one ParA dimer, preventing random diffusion (Surovtsev et al, PNAS 2016). The revised manuscript has been revised to include this discussion (lines 399-417).

Regarding the drop in the intergenic sequence for both ParB and ParZ, this was an artefact of using the wild-type genome as the reference genome for ChIP-seq read mapping; the real genomes of these strains include a *P_{flaB}-kan* cassette in the *parB-parZ* intergenic region, which could not be mapped to the wild-type genome. We have addressed this issue by remapping the data on the genome of the engineered strains used for the experiment. See lines 708-734. We have revised all of our ChIP-seq figures accordingly, and updated the data files deposited to GEO.

- Data in Supplementary figure 6E (ChIP-seq profile of ParZ with *parZ* on a shuttle plasmid) is not so convincing to show that the binding site of ParZ is within the *parZ* gene. There are several peaks and

summits on that profile, and the peak within *parZ* is not the most prominent one. A control of anti-ParZ ChIP-seq on an empty plasmid is perhaps required. It would be very convincing if Takacs et al could inspect the sequence immediately below the summit to identify any potential inverted repeat. No further experiment is required but some more analysis would strengthen the claim and add to the novelty of the ParZ protein.

Response: Thank you for bringing this to our attention. The original supplementary figure 6e was actually underestimating the number of reads for the *parZ* sequence on the shuttle vector. This is because the program was randomly assigning *parZ* reads between the chromosomal and plasmid *parZ* sequence, leading to a 50:50 distribution. This was incorrect because there are about 5 times more shuttle vectors than chromosome copies (see Kasumba et al 2015, ref 22; Tilly et al 2006, ref 23, and Beurepaire and Chaconas, ref 63). When 83% of the *parZ* reads are assigned to the shuttle vector (to reflect its 5-fold higher copy number than the chromosome), the ChIP-seq peak within the *parZ* region on the shuttle vector becomes the predominant one, which is now shown in Supplementary Fig. 4c-e. To provide further support for our hypothesis, we showed that ParZ-msfGFP form many more fluorescent foci when expressed from the shuttle vector compared to a strain carrying chromosomally-encoded *parZ-msfGFP* and an empty vector (Supplementary Fig 4f-g). Together, these ChIP-seq and imaging data support the notion that ParZ binds its own sequence. We also found both direct (empty arrowheads) and inverted (filled arrowhead) repeats in the *parZ* sequence (see below). However, we agree with the reviewer that at this stage, this remains a hypothesis. We have therefore revised the text to tone our claim down to a suggestion. See lines 266-277 in the revised text.

- Discuss the Myxococcus PadC-ParABS-SMC system if appropriate. This is another system that shows

the plasticity in bacterial chromosome segregation.

Response: The PadC-ParABS-SMC is indeed an interesting system to discuss in the context of our story. This is now done. See lines 392-395. Thanks for the suggestion.

- Table S2: indicate the antibodies used for ChIP-seq, anti-GFP/mCherry or anti-ParZ/ParB/SMC polyclonal antibodies etc.

Response: Good point. This information is now included. Thanks!

Reviewer #1 (Remarks to the Author):

The authors have adequately addressed all questions and concerns, and this reviewer has no additional comments.

Reviewer #2 (Remarks to the Author):

Authors have addressed all my comments - congratulations on this excellent work.

Reviewer #3 (Remarks to the Author):

I am happy with the revised manuscript, and have no further comment. Thank you for an opportunity to review for Nat Comms.